# Comprehensive financial health assessment using Advanced machine learning techniques: Evidence based on private companies listed on ChiNext

**Wen Zhu, Meiling Li** ◉ *, **Chengcheng Wu** ◉ *, **Shanqiu Liu**

Guangzhou Huashang College, Guangzhou, Guangdong, China

◉ These authors contributed equally to this work.
* 9524761@gmail.com (ML); celia.cheng@163.com (CW)

**Data Availability Statement:** All relevant data are within the manuscript and its Supporting Information files.

## Abstract

This study develops a specific and measurable framework for assessing the financial health (FH) of privately-owned companies listed on ChiNext, aimed at identifying financially sound enterprises and helping investors avoid losses caused by financial fraud or earnings management. The research proposes and tests four hypotheses related to key financial indicators and one overarching hypothesis regarding the model's performance. Using gradient boosting machines and random forests, the model achieves high accuracy and robustness against overfitting through iterative learning. The framework incorporates four pairs of financial indicators and two non-financial indicators into four classifiers, significantly outperforming the Altman Z-score model in predicting financial soundness. Among 75 private companies with special treatment by the Securities Regulatory Commission in Shanghai and Shenzhen in 2022, 72 were correctly identified as sub-healthy or unhealthy, achieving an accuracy rate of 96%. This study demonstrates time-bound practical value by validating the model with 2022 data and highlights its relevance for cross-market applications. The results provide achievable solutions for enterprise managers and policymakers in financial decision-making and risk management.

## 1. Introduction

Idea of financial health can be seen as the ability of the company to maintain a balance against changing conditions of the environment and at the same time in relation to everyone participating in the business [1]. In the field of modern finance, the precise measurement of financial health holds paramount importance. It not only pertains to core financial activities such as company earnings or profit management but also encompasses the decision-making process of enterprises when they encounter financial difficulties. Generally, business failure brings about severe economic and social consequences. Hence, the matter of financial health should be treated with great seriousness [2]. With the increasing complexity of global financial markets, accurate assessment of corporate financial health has become a key factor in ensuring

**Funding:** This study was supported by the 2024 Major Research Project at Guangzhou Huashang College, "Exploration of Theory and Practice of Financial Health of Listed Companies Empowered by AI" [Grant Number: 2024HSZD02 to W.Z.]; the Key Discipline Research Capacity Enhancement Project of the Guangdong Provincial Department of Education, 2021 [Grant Number: 2021ZDJS133 to M.L.]; the Guangdong Provincial Undergraduate Teaching Quality and Teaching Reform Project, 2022 [Grant Number: 263 to M.L.]; and the 2020 Institutional Key Discipline Funding (Second Batch) [Grant Number: 114 to M.L.]. No additional external funding was received for this study.

**Competing interests:** The authors have declared that no competing interests exist.

market stability and promoting economic growth. The work of Netemeyer et al.[3] emphasises the close link between individuals' perceptions of financial health and their overall well-being, revealing the major importance of financial health assessment for individual and corporate decision-making. Furthermore, the Financial Health Network report published in 2022 points out that diagnosing and monitoring financial health can provide organisations with crucial insights, assisting them in making wiser decisions in the ever-changing market environment.

Financial ratios have traditionally been indicators of a corporate overall performance and may help quantify a potential impact of internal ratings on financial performance [4]. But when discussing the theoretical framework for assessing corporate financial health, relying solely on surface data from annual financial statements for analysis often provides a limited perspective. For instance, by analysing profit and loss statements, we can observe trends of profit increase or decrease between consecutive financial years, but it is difficult to deeply analyse the reasonableness, stability, or implied "true value" of profit growth, so potential underlying factors are overlooked [5]. In fact, the sustained improvement or deterioration of a company's profitability often result from multiple factors interacting and accumulating, rather than a short-term achievable goal [6]. For example, a company's sustained profit growth may stem from their management's efforts over the years related to cost control, revenue growth, tax strategies, and financial strategies. Conversely, a sustained decline in profits may be caused by various factors such as a deteriorating macroeconomic environment or poor management. Among Chinese listed companies in the capital market, management teams often hesitate to face the reality of significant profit declines, fearing the impacts these might have on stock prices, regulatory warnings, financing capabilities, and incentive plans. Management might then take a series of seemingly reasonable measures to adjust profit levels, sometimes even resorting to financial fraud.

This study addresses the research question: How can machine learning models enhance the effectiveness of profit health assessment by integrating both financial and non-financial indicators? While existing research primarily focuses on financial fraud detection and earnings manipulation, there is a gap in providing comprehensive profit health assessments. The absence of methods combining financial and non-financial indicators limits the ability of investors and stakeholders to quickly and accurately identify companies with healthy financial conditions. To address this gap, this study builds on theories of earnings management, financial distress, and signal transmission to construct a profit health assessment framework. This model aims to overcome the limitations of traditional financial ratios by using a hybrid approach that integrates financial and non-financial data. Furthermore, this study endeavours to combine this model with the Z-score model to enhance the effective identification of and ability to classify corporate financial health status [7], which offers an important assessment tool for financial conditions and risks to corporate stakeholders.

The article introduces, analyzes empirically, and applies the financial health model through the following chapters:

- **Introduction**: Establishes the context, research questions, and research gap.

- **Literature Review**: Summarizes key theories and previous studies.

- **Theoretical Framework and Hypotheses**: Proposes research hypotheses and conceptual models.

- **Methodology**: Details the research methods and data collection strategies.

- **Empirical Research**: Variable definition and model construction

- **Empirical Results and Robustness Test**: Analyzes the empirical results and ensures their stability.

- **Aplication and Expansion**: Application of profit health evaluation and extend to financial assessment

- **Conclusion**: Highlights the contributions and practical implications of the study.

## 2. Literature review

Over the past three decades, academic research on financial health has primarily focused on the continuous deepening of, and innovations in, earnings management behaviours. The research outcomes mainly centre around false revenue recognition [6], inappropriate expense deferral or capitalization [5] malicious accounting estimates and judgements [8], profit manipulation through reserve manipulation [9], false asset and liability valuation [10], related-party transaction manipulation [11], and other unethical practices [12]. Through theoretical analysis, case studies, and empirical research related to motivations, implementation methods, and the impacts of these behaviours, this literature has not only deepened our understanding of earnings manipulation but also provided practical guidance for identifying and addressing manipulative behaviours in financial reporting. These works have also laid the foundation for exploring research gaps related to financial health.

### 2.1 Information asymmetry leaves room for profit manipulation

Earnings management, defined as interventions in the financial reporting process by enterprises to meet certain individual or collective interests, is also regarded as misleading stakeholders about the actual situation of an enterprise through accounting or business manipulation. In the context of research into China's A-share ChiNext, the measurement of financial health has assumed a critical role due to the prevalent information asymmetry between shareholders and management. The decision-making behaviours of management, particularly during financial distress, exert substantial impacts on the financial health of enterprises. Sumiyana et al delved into the investment decision-making behaviours of CEO in the face of financial distress and scrutinised how earnings management influences these decisions, underscoring the profound effect of management decisions on an enterprise's financial condition [13]. Yao Hong et al posited that information asymmetry prevails not only between shareholders and management but also among other external information users and management [14]. This asymmetry facilitates profit manipulation and earnings management, rendering it challenging for investors and other corporate stakeholders to accurately predict whether a company will enter financial distress based on financial indicators, potentially leading to investment missteps. ALJAWAHERI discovered that earnings manipulation adversely affects investors' behaviour in the financial market, given its long-term detrimental effects on investor behaviour and the reliability of financial statements [15].

Despite numerous similarities in the profit manipulation methods employed by enterprises internationally and those in mainland China, the tactics utilised by Chinese firms have been noted to be more varied, intricate, and covert [16]. Earnings management is particularly widespread post-IPO to enhance stock performance and alleviate financing pressure [17]. Numerous listed companies - especially those already incurring losses - resort to earnings management with the objective of preserving their listing status, with firms often engaging in earnings management to artificially inflate profits [18]. Earnings management can be bifurcated into accrual-based and real earnings management; the former involves manipulating

profits by adjusting financial statement data, whilst the latter regulates profit through actual business operations. Notably, while non-standard audit opinions can deter accrual-based earnings management, they may encourage real earnings management, which is difficult to detect due to its concealed nature. This suggests that real earnings management activities might still transpire under standard audit opinions [19]. Hence, research into financial health is especially pivotal as it aids investors in identifying genuinely financially healthy enterprises.

## 2.2 Financial warning studies provide references for building theoretical models

Signal transmission theory emphasises that abnormal fluctuations in the financial indicators of enterprises may imply the existence of financial fraud or embellishment behaviours [5]. Financial warning models utilise this theory to predict the likelihood that enterprises will face financial crises by analysing the correlation between enterprise data and financial risks. This provides important warning information with which enterprise management teams can adjust their business strategies in a timely manner and avoid potential bankruptcy or default risks. It also offers important references for investor decision-making so that investment risks can be avoided [7].

Although financial warning models play important roles in predicting corporate financial risks, they face the problem of collinearity among some indicators, leading to excessive redundancy of information and affecting the model's discriminant efficiency [6]. In addition, most existing models adopt static modelling, which is difficult to adapt to the complex and changeable internal and external environments of enterprises. This static approach may misjudge some enterprises as being in a "grey area", making the evaluation of their financial condition vague and making it difficult to provide accurate information to investors [20].

In the context of improving model accuracy and adaptability, Xiao et al further emphasised that the operating conditions of enterprises can be revealed by analysing various financial and non-financial data, effectively mining potential crisis signals [21]. Yin and Yang believe that using financial indicators for financial crisis prediction is not only necessary but also reasonable [22]. However, Liu et al. proposed that although the choice of financial indicators directly affects the accuracy of financial warning models, there is currently no unified theoretical framework or mature theoretical support for selecting financial indicators [23]. Wang and Bai emphasised the importance of comprehensively considering factors such as enterprise operating conditions, industry characteristics, and product life cycles [24]. Although these attempts help optimise the variable selection and improve the predictive ability of the model, they may overlook the intrinsic signal transmission mechanism between financial indicators and their manifestation in the financial condition of enterprises. For example, frequent changes in accounting firms or asynchronous changes in financial indicators may violate normal signal transmission mechanisms, thereby revealing potential financial problems.

This comprehensive perspective indicates that improving the accuracy and practicality of financial warning models requires in-depth improvements to and refinements of the existing methods. Exploring new theoretical frameworks and technical methods such as machine learning and big data analysis is expected to solve the above challenges and improve the effectiveness of financial warning models.

## 2.3 Signal transmission studies provide inspiration for building theoretical models

Li et al advocated for a comprehensive consideration of the interconnections between various accounting items and the consistency of financial statements as crucial for developing an

integrated indicator system for profit manipulation detection, highlighting abnormal indicator characteristics and their potential implications [25]. For instance, the ratio of the growth rate of accounts receivable to the growth rate of sales revenue, whether excessively high or low, could be deemed a potential warning signal. Beneish also indicated that the likelihood of profit manipulation would escalate with anomalies in accounts receivable [26].

You highlighted that a company's profitability and cash flow capability stand as the most sensitive indicators of its financial condition, directly reflecting its financial health and being critical factors that could precipitate a financial crisis [27]. Through analysing the consistency between accounting profits and net cash flow, Wu evaluated whether companies engage in accounting earnings management behaviours due to cash flow shortages [28]. Xu et al. suggested the use of the ratio of net cash flow from operating activities to total profit or net profit as a metric for detecting profit manipulation behaviour [29]. Yang and Zhang underscored that real earnings management entails management's alteration of decisions concerning business operations, which directly impacts cash flow from operating activities, predominantly sourced from sales revenue [30].

Moreover, in the development of a Bayesian financial forecasting model based on composite attributes, Meng et al selected forecast indicators including main business revenue and the inventory turnover rate [31]. Cheng, in constructing his incremental model, likewise incorporated key indicators such as the accounts receivable turnover rate, inventory turnover rate, and growth rate of main business revenue [20]. Kondo et al, in their study on identifying and predicting accounting fraud, also chose variables including accounts receivable, earnings, cash flow, inventory, and fixed assets, clearly emphasising the significance of inventory indicators [32].

Azam et al pointed out that companies might manipulate period costs for financial fraud, highlighting the complexity of cost and expense management and the key role this plays in maintaining financial health [33]. Theoretically, a growth in gross revenue should directly promote an increase in company profits. However, if the gross revenue and total profit trends change inconsistently, this might reflect the company's improper management of costs and period expenses, such as excessive consumption or inefficiency, meaning they have failed to effectively convert into profit growth. Moreover, this phenomenon might also be a sign of profit inflation by the company, indicating potential issues with their financial condition.

This study was inspired by the aforementioned previous research on signal transmission between financial indicators, revealing various potential directions for improvement in this field to further expand the depth and breadth of the related research. It should be emphasised that the impact of earnings management on the long-term value and sustainable development of enterprises still requires further research [5]. This study aims to bridge the gaps in the existing literature by quantifying potential earnings management behaviours, as well as medium- to long-term profit health anomaly signals based on non-financial indicators, using the visualisation techniques of machine learning algorithms [34]. This will reveal the inherent correlation and change mechanism between different financial signals, and, combined with the Z-score model, will provide an assessment method for financial health. This method has a certain universality and provides a new perspective for cross-cultural and cross-border research on financial health. This will develop our understanding of the differences and similarities in earnings management behaviours in different regulatory environments and cultural backgrounds [35,36].

## 3. Theoretical framework and research hypotheses

The aim of this study was not to focus on the positive aspects of earnings management by companies, nor does it solely concentrate on the specific means of profit manipulation. Instead, it focuses on the abnormal signals in the medium- to long-term profit variations of companies.

These signals reveal the overall earnings management behaviours that companies might adopt. By analysing the correlation between indicators, this study explored the construction of several detectors aimed at uncovering traces of profit manipulation and earnings management in order to assess the financial health status of companies. Inspired by Maslow's hierarchy of needs theory and the balanced scorecard theory proposed by Kaplan et al, this study proposes an innovative framework for evaluating profit health [37]. This framework is divided into four levels, with the fourth level representing the highest level of profit health and the first level indicating the lowest level. The principle behind this framework is to examine any company's performance over a continuous five-year period, calculate the annual scores of four detectors and two non-financial indicators, add the scores for each year to obtain a five-year total profit health score, and finally, based on the different total scores, stratify the profit health.

### 3.1 Theoretical framework

This theoretical model adopts a unique olive-shaped structure design, symbolising that the levels representing the highest and lowest profit health proportions in the overall distribution of profit health are relatively small, while the middle two levels occupy larger proportions. The olive-shaped model shown in Fig 1 includes four detectors, $I_1$ to $I_4$, internally, and two non-financial indicator detectors, Changes in the Number of Accounting Firm Engagements (*NCAF*), and Audit Opinion (*AO*), externally, to comprehensively measure the profit health of companies.

Detector $I_1$ analyses the accounts receivable growth rate (*ARGR*) and revenue growth rate (*RGR*) to identify potential anomalies in revenue growth. According to basic accounting principles, if revenue growth is not accompanied by a corresponding increase in accounts receivable, it is marked as an anomaly and scores 1 point; otherwise, it scores 0 points. Detector $I_2$ combines the revenue growth rate (*RGR*) with the operating cash flow growth rate (*NCFGR*) to evaluate the true value of revenue growth. If revenue growth is inconsistent with the growth in net cash flow from operating activities, it is considered an anomaly and scores 1 point; otherwise, it scores 0 points. Detector $I_3$ focuses on the relationship between the net profit growth rate (*NPGR*) and the inventory turnover rate growth rate (*ITR*) to judge the reasonableness of

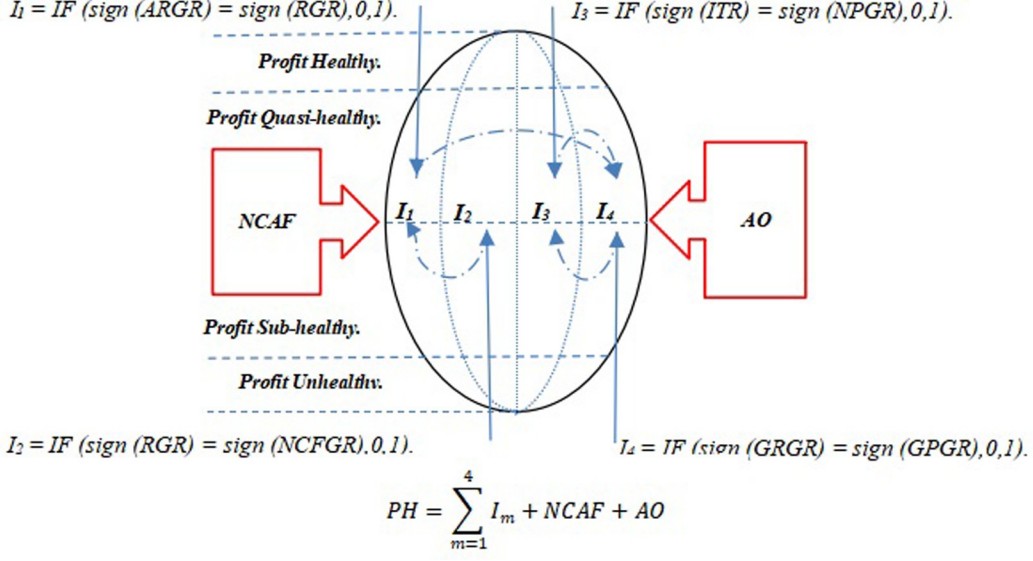

**Fig 1. Theoretical model of profit health assessment.**

net profit growth. According to accounting theory, an increase in the inventory turnover rate should promote profit growth, so if this rate is not coordinated with net profit growth, it is considered an anomaly and scores 1 point; otherwise, it scores 0 points. Detector $I_4$ analyses the gross profit growth rate (*GPGR*) and gross revenue growth rate (*GRGR*) to detect signs of manipulation of gross profit. If gross revenue growth is accompanied by a decrease in gross profit, or if gross profit increases while gross revenue decreases, both are considered abnormal phenomena and score 1 point; otherwise, they score 0 points.

Although these four detectors are independent in scoring, they form a closely related logical chain. In detector $I_1$, a growth of accounts receivable directly affects the truthfulness of revenue growth, which further affects the increase in gross income in detector $I_4$, thereby driving the increase in total profit. At the same time, an increase in total profit affects the growth of net profit in detector $I_3$. To verify the truthfulness of net profit growth, it is necessary to check whether the direction of net profit growth is consistent with the direction of inventory turnover rate growth in $I_3$. The consistency between the growth direction of net cash flow from operating activities in detector $I_2$ and the revenue growth direction is used to evaluate the revenue realisation capability. This series of checks not only interlocks but also collectively constitutes the basis for in-depth analysis of the financial health status of companies.

Furthermore, while the financial indicators obtained and calculated from publicly available annual reports are objective, they may not be comprehensive. Therefore, this model considers incorporating management behaviour—the frequency with which accounting firms changed within the last five years (*NCAF*, scoring 1 point for each change)—and combining the audit opinions given by accounting firms in the annual reports over the past five years (*AO*, scoring 0 points for "unqualified opinions", 1 point for "qualified opinions with explanatory paragraphs", 3 points for "qualified opinions", and 5 points for "adverse opinions" or "disclaimer of opinion").

Over the five-year period, the total profit health (*PH*) score of the company is calculated using the scores of the four financial indicator detectors and two non-financial indicators. By analysing the correlation between the indicators, four detectors were constructed to uncover traces of profit manipulation and earnings management, thereby evaluating the financial health status of companies. Subsequently, this study addresses two key issues: first, validating the effectiveness of the selected financial and non-financial indicators and four detectors in quantifying profit health in a big data model; second, based on the empirical results for profit health, using an equal interval grouping method to determine the range of four levels of profit health. The profit health evaluation theoretical model proposed in this study aims to provide insights for securities market regulatory agencies and policymakers, as well as enhance the ability of investors and auditors in the accounting and auditing fields to identify and respond to financial manipulation, as demonstrated by the studies of Cohen et al. [38] and Nelson et al. [39].

### 3.2 Research hypotheses

This study constructs a detection indicator $I_1$, composed of the growth rates of accounts receivable and operating revenue, drawing inspiration from the related research by Li et al. [25] and Beneish [26]. Generally, the growth of accounts receivable should align with that of operating revenue, meaning the directions of growth should be congruent. However, in instances where the directions of growth diverge, further analysis is warranted: firstly, if the growth rate of accounts receivable is negative while that of operating revenue is positive, this may reflect, to some extent, efforts by the management to enhance collection efficiency or to implement advance payment sales policies. Although this scenario might seem beneficial for the company at first glance, considering the downward trend in China's macroeconomic

environment in recent years, reliance on cash sales models or excessive dependence on advance payment sales could adversely affect the company's cash flow, especially when the external economic environment deteriorates further. Secondly, a positive growth rate in accounts receivable coupled with a negative growth rate in operating revenue is not easily interpreted as a positive signal and is more likely viewed as a negative indicator, such as a decline in market demand or diminished collection capabilities. Additionally, this situation may involve the manipulation of financial statements, for example, companies might overly aggressively recognize revenue to beautify financial reports, especially against a backdrop of declining sales revenue, by artificially increasing accounts receivable to inflate reported operating revenue; or through fabricating sales or "round-tripping" sales, artificially increasing both accounts receivable and operating revenue to conceal the true operational state of the business. Hence, hypothesis $H_1$ is proposed:

$H_1$: The relationship between the growth rate of sales revenue and the growth rate of accounts receivable significantly affects the profit health of a company, and inconsistent trends in these areas may signal unhealthy profits.

The detection indicator $I_2$, constituted by the growth rates of operating revenue and the net cash flow from operating activities, references the research findings of You [27] and Yang and Zhang [30]. Typically, when the growth directions of these two indicators align, it reflects a relatively healthy operating condition for the enterprise. However, when these indicators exhibit divergent growth directions, a detailed analysis is required: (1) A positive growth rate in operating revenue coupled with a negative growth rate in net cash flow from operating activities may, from an optimistic perspective, indicate that the company has consumed more cash in the short term due to prior investment actions, which are expected to yield long-term benefits and growth in operating revenue. This situation could also arise from the company sacrificing cash flow to increase accounts receivable, or due to seasonal factors (such as ramping up production before a festival to boost sales, followed by post-festival cash receipts). Yet, from a pessimistic viewpoint, this could signify the company is experiencing cash flow constraints, and the practice of temporarily boosting operating revenue through excessive credit sales or reliance on other non-sustainable income sources is unsustainable; (2) When the growth rate in operating revenue is negative while the net cash flow from operating activities is positive, from a positive angle, this could reflect that management is optimising cash flow management or there has been an increase in cash flow from non-operating activities. However, from a negative perspective, this situation might signal an increasing risk of declining revenue, which could affect the company's profitability in the long run. Additionally, it could mean that the company is masking the deterioration of its actual operating condition by recognising unconventional cash gains—such as proceeds from asset sales. Hence, hypothesis $H_2$ is proposed:

$H_2$: The relationship between the growth rate of sales revenue and the growth rate of net cash flow from operating activities significantly affects the profit health of a company, and inconsistent trends in these areas may signal unhealthy profits.

The detection indicator $I_3$, composed of the growth rate of net profit and inventory turnover ratio, bases its theoretical framework on the studies by Cheng [20] and Kondo et al. [32]. Theoretically, an improvement in the inventory turnover ratio should facilitate an increase in net profit, making a congruent direction of growth between the two a foreseeable occurrence. However, in cases where their growth directions diverge, two scenarios are discussed: (1) When the growth rate of net profit is positive, but the inventory turnover ratio is negative, from a positive viewpoint, this could signify that management is adjusting marketing strategies (such as increasing product prices) or effectively implementing cost management measures.

Although the reliability of such positive interpretations might be questioned in a global context of overcapacity and accelerated product updates, interpreting it as a negative signal appears more reasonable. Negatively, this situation could result from the company excessively accumulating inventory, leading to impaired liquidity and long-term profitability, or a mismatch between products and market demand (e.g., an increase in inventory without a corresponding increase in sales), which could harm the company's market position and profitability in the long term; (2) When the growth rate of net profit is negative, and the inventory turnover ratio is positive, from a positive angle, this could reflect the company optimising inventory management or increasing investment in the short term (such as allocating funds for the development of new products or market expansion activities, thereby causing a short-term decline in net profit but an improvement in inventory turnover). However, from a negative perspective, this might indicate issues with the sales strategy (such as stimulating sales through excessive price reductions, leading to a decline in profit margins) or a decrease in revenue quality (the reduction in net profit could reflect a decline in revenue quality or an increase in costs). Despite an increase in inventory turnover, the inability to effectively translate this into sufficient profits necessitates the company to reassess its product pricing strategy and cost structure. Hence, hypothesis $H_3$ is proposed:

$H_3$: The relationship between the net profit growth rate and inventory turnover rate growth rate significantly affects the profit health of a company. When inconsistent trends are exhibited in these two areas, it may indicate unhealthy profit signals.

For Detector $I_4$, it is constituted by the growth rate of gross revenue and the growth rate of total profit [33]. Evidently, a positive growth rate in gross revenue tends to facilitate the growth of total profit, making their concurrent growth direction easily explainable. When the growth directions are opposite, two scenarios warrant discussion: (1) A positive growth rate in gross revenue accompanied by a negative growth rate in total profit might, from an optimistic viewpoint, be indicative of an expansion in market investment scale, product development and innovation, or optimisation of the supply chain and production capabilities. Conversely, from a pessimistic viewpoint, it might be interpreted as improper cost control or an overreliance on low-profit products; (2) A negative growth rate in gross revenue alongside a positive growth rate in total profit could, from a positive perspective, suggest improvements in cost and efficiency or an increase in sales of high-profit products. Given the intense market competition currently, interpreting this phenomenon as a positive signal may not be highly reliable. However, from a negative standpoint, it could be interpreted as an increase in one-off gains rather than a stable increase in profits from core business activities, potentially masking a deterioration in the actual operating condition, or even representing potential financial statement manipulation. For instance, a company might, through accounting practices such as the excessive capitalisation of certain expenses, delay or conceal actual operating costs, artificially inflating the total profit. Such practices could mislead investors and shareholders. Hence, hypothesis $H_4$ is proposed:

$H_4$: The relationship between the growth rate of total profit and the growth rate of gross revenue significantly affects the profit health of a company. When inconsistent trends are exhibited in these two areas, it may indicate unhealthy profit signals.

Relying solely on a single financial ratio as a predictive indicator of a company's financial health has inherent limitations, so incorporating non-financial indicators becomes particularly important. Muñoz-Izquierdo et al revealed that models integrating financial and audit information significantly outperform those relying solely on financial ratios in terms of prediction accuracy [40]. This finding highlights the higher predictive value of audit information over

traditional financial statement data in forecasting corporate bankruptcy risk. And the information in audit fees can be used to provide an alternative measure of a firm's accounting quality [41]. Zhou also noted that non-financial indicators, such as non-standard audit opinions and changes in accounting firms, could also imply the existence of financial statement fraud [42].

Based on these research findings, this paper incorporated two non-financial indicators as control variables: types of audit opinions and the number of changes in accounting firms. These two non-financial indicators, along with the previously mentioned four pairs of financial growth rate indicators—the growth rate of accounts receivable (*RGR*) and growth rate of sales revenue (*ARGR*); the growth rate of sales revenue and net cash flow from operating activities growth rate (*NCFGR*); the inventory turnover rate growth rate (*ITR*) and net profit growth rate (*NPGR*); and the total profit growth rate (*GPGR*) and gross revenue growth rate (*GRGR*)—and seven financial indicators, collectively constitute a comprehensive evaluation framework for assessing the profit health of enterprises. Accordingly, the overarching model hypothesis *H* is proposed.

*H*: The growth rate of accounts receivable, the growth rate of sales revenue, the net cash flow from operating activities growth rate, the inventory turnover rate growth rate, the net profit growth rate, the total profit growth rate, the gross revenue growth rate, types of audit opinions, and the number of changes in accounting firms influence profit health.

## 4. Methodology

In recent years, the random forest algorithm has attracted substantial attention from scholars due to its high accuracy and rapid learning ability, while its performance is unaffected by the nature of the dataset, offering a clear advantage over multivariate linear regression models that require normally distributed data. Mehrani et al. improved the accuracy with which profit manipulation could be predicted by combining financial and non-financial ratios, using traditional logistic regression models and metaheuristic models based on neural networks and genetic programming [43]. Bertomeu et al. [44] uses a wide set of variables from accounting, capital markets, governance, and auditing datasets to detect material misstatements by implying machine learning. Ding noted that the random forest algorithm significantly improves prediction model performance compared to existing algorithms such as bagging, dagging, Adaboost, Multiboost, and the random subspace method [45]. Ye et al found that the random forest performed excellently against various other algorithms—including artificial neural networks (ANN), logistic regression (LR), support vector machines (SVM), CART, decision trees, Bayesian networks, bagging, stacking, and Adaboost—in exploring financial statement fraud detection [46]. Patel successfully applied the random forest model to detect financial statement manipulation among Indian listed companies [47]. Whiting have used machine learning methods such as random forests, gradient boosting, and rule ensembles, selecting indicators like the growth rate of accounts receivable, inventory growth rate, cash flow, and gross margin ratio to predict accounting fraud [48]. Rahman have found the random forest model to be effective in predicting earnings manipulation among Malaysian listed companies [49].

In view of the above research results, since some indicators may have strong collinearity, this study consider using nonlinear regression models to test research hypotheses. Due to its high predictive accuracy and strong explanatory power, as well as its robustness regarding missing values and outliers, the gradient boosting machine algorithm was chosen. So this study first verified whether 7 financial indicators and 2 non-financial indicators have a significant impact on profit health by constructing a gradient boosting machine model; due to the lack of requirements for data distribution and data type; its good tolerance for noise and

outliers; and the advantage of visualising the partial dependence plot of each variable's influence on profit health, the random forest algorithm was chosen. Secondly, based on the intrinsic correlation of these indicators, these indicators were decomposed and combined into 4 detectors, and test the explanatory power of each recognizer on *PH* based on the random forest model. In addition, by integrating the traditional Z-point financial crisis early warning model, we innovate and expand the limitations of the traditional financial early warning model, and strive to achieve accurate predictions of the financial health of enterprises in a dynamic and complex market environment. This constitutes the composition of this paper. an important theoretical contribution.

## 5. Empirical research

### 5.1 Sample selection and data collection

This study takes private companies listed on the ChiNext board of A-shares in China as the research object and collects financial and non-financial data from 2018 to 2022 from the Wind database. As of December 31, 2018, there were 580 private enterprises listed on the ChiNext board (excluding ST or *ST), with a total of 2,900 records. After removing outliers, 2,622 valid records remain. ST and *ST companies are removed from the sample set to reduce model testing bias due to sample imbalance. Extreme values of various indicators are also removed. These steps ensure research rigor and result reliability, providing a solid data foundation for in-depth analysis of the financial health of private listed companies on the ChiNext board.

### 5.2 Variable definitions

This paper employs *ARGR*, *RGR*, *NCFGR*, *NPGR*, *ITR*, *GPGR*, and *GRGR* as independent variables; *AO* and *NCAF* as control variables; and profit health as the dependent variable, as delineated in Table 1. Taking *ARGR* as an example, it illustrates the calculation of the accounts receivable growth rate for a given year, where the accounts receivable growth rate for year T = (accounts receivable in year T minus accounts receivable in year T-1) / absolute value of accounts receivable in year T-1. The calculation methods for the other financial indicator growth rates were the same as for the accounts receivable growth rate.

All the above variables are dimensionless variables. Among them, *PH* is the score calculated by each recognizer. *ARGR*, *RGR*, *NCFGR*, *NPGR*, *ITR*, *GPGR*, and *GRGR* are the calculated relevant growth rates. *AO* is the score assigned according to different audit opinion types. *NCAF* is the number of times an enterprise changes accounting firms, with one point for each change.

**Table 1. Variable definitions and data sources.**

| Variable Type | Variable Name | Explanation | Data Source |
|---|---|---|---|
| Dependent Variable | *PH* | Profit Health (score) | Calculated based on $I_1$-$I_4$ |
| Independent Variables | *ARGR* | Account Receivable Growth Rate | WIND database (2018–2022) |
| Independent Variables | *RGR* | Revenue Growth Rate | |
| Independent Variables | *NCFGR* | Net Cash Flow from Operating Activities Growth Rate | |
| Independent Variables | *NPGR* | Net Profit Growth Rate | |
| Independent Variables | *ITR* | Inventory Turnover Ratio Growth Rate | |
| Independent Variables | *GPGR* | Gross Profit Growth Rate | |
| Independent Variables | *GRGR* | Gross Revenue Growth Rate | |
| Independent Variables | *AO* | Audit Opinion Type Score | |
| Independent Variables | *NCAF* | Number of Changes in Accounting Firm | |

## 5.3 Descriptive statistics

Based on the panel data set, a scatter line chart is drawn based on the annual sign similarity and difference counts of each pair of variables for each identifier and the annual mean of each pair of indicators. The x-axis represents five fiscal years, the left scale shows the count of similar and different signals (signs) for that pair of indicators each year, and the right scale shows the magnitude of the average values for that pair of variables annually. In Fig 2, green dots indicate the number of times each pair of variables moved in the same direction in that year, while red crosses signify the number of times each pair of variables moved in opposite directions. For ease of description, the images in Fig 2 are respectively labeled as *(a)*, *(b)*, *(c)*, and *(d)* from top to bottom.

In Fig 2(A) presents the scatter line graph for Identifier 1, illustrating the relationship between the accounts receivable growth rate (*ARGR*) and the sales revenue growth rate (*RGR*). It can be observed that the mean value trends of *ARGR* and *RGR* are broadly parallel from 2018 to 2022, mirroring the co-directional increase or decrease relationship between accounts receivable and sales revenue in accounting practices. Typically, the directions of change for these two factors are congruent. Nonetheless, within the panel dataset, instances of divergent *ARGR* and *RGR* trends are noted. For instance, in 2018, the instances of concordant *ARGR* and *RGR* signs were approximately 400, while the instances of discordant signs were around 100. Such a discrepancy could signify potentially unhealthy profit conditions in a business.

The graph in Fig 2(B) is the scatter line graph for Identifier 2, showcasing the relationship between the sales revenue growth rate (*RGR*) and the net cash flow from operating activities

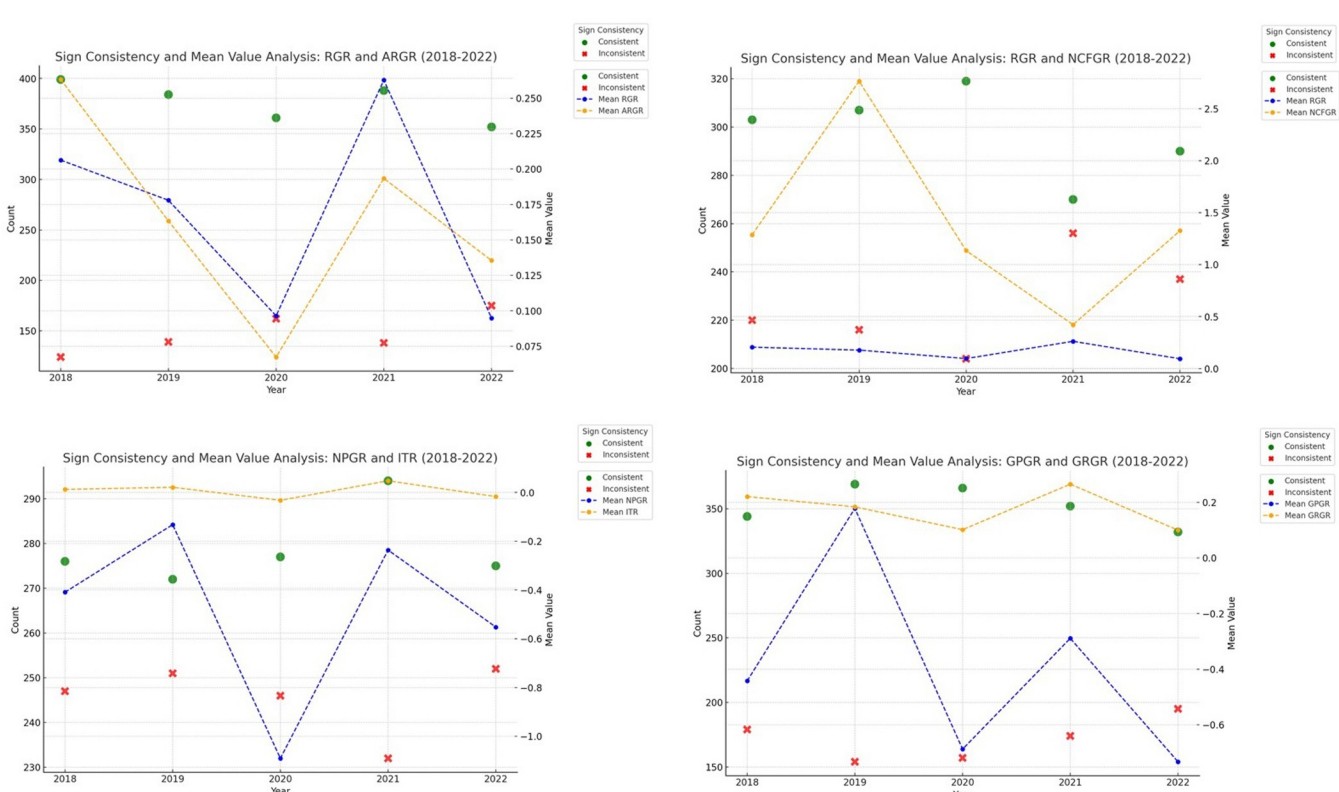

**Fig 2. Scatter line graph of the number of similar and divergent signals for each pair of financial indicators and Mean values for each pair of indicators annually.**

growth rate (*NCFGR*). Fig 2(B) reveals that the variance in *NCFGR* over the five years from 2018 to 2022 is considerably greater than that for *RGR*, potentially reflecting the overall instability of cash flow in ChiNext companies or the occurrence of significant events in specific years. Notably, 2021 saw a significant increase in *RGR* alongside a significant decrease in *NCFGR*, marking the year with the highest number of instances where the two financial indicators moved in opposite directions. This may have been a consequence of the COVID-19 pandemic's impact in 2020, which led to a poor cash recovery. Overall, the instances where the signs are identical surpass the instances of divergence, suggesting that *RGR* and *NCFGR* should typically exhibit consistent signs. A discrepancy would likely emit signals of unhealthy profits.

The graph in Fig 2(C) illustrates the changes in characteristics between the net profit growth rate (*NPGR*) and the inventory turnover rate growth rate (*ITR*) for Identifier 3. This graph unveils that the variation in *NPGR* substantially exceeds that of *ITR*. Specifically, the notable decline in *NPGR* in 2020 may reflect the repercussions of significant events, such as the global pandemic, on company profitability. Such pronounced volatility could signify that businesses encountered heightened financial risks, necessitating various strategies to mitigate these risks, including cost control measures. Compared to *NPGR*, *ITR* demonstrates less volatility, possibly indicating cautious inventory management by firms during the pandemic period. However, the instances where the signs are identical always surpass the instances where they differ, implying that *NPGR* and *ITR* should ordinarily exhibit parallel movements. A discrepancy would suggest that inventory turnover has not resulted in concurrent net profit growth, potentially flagging unhealthy profits.

The graph in Fig 2(D) depicts the changes in characteristics between the gross revenue growth rate (*GRGR*) and the total profit growth rate (*GPGR*) for Identifier 4. This graph shows that the mean values of *GPGR* undergo considerable fluctuations across different years. Such variability might be attributed to market and operational factors, such as unstable sales revenue, alterations in cost management, or shifts in macroeconomic conditions. Conversely, the mean values of *GRGR* are comparatively stable, indicating less volatility from 2018 to 2022. Nonetheless, the instances where the signs are congruent consistently outnumber the instances where they differ, suggesting that *GRGR* and *GPGR* should typically trend in the same direction. A discrepancy could act as an early warning signal of unhealthy profits.

## 5.4 Correlation analysis

This study takes *ARGR*, *RGR*, *NCFGR*, *ITR*, *NPGR*, *GPGR*, *GRGR*, *AO*, and *NCAF* as independent variables, with PH as the dependent variable. Using panel data from China's A-share Growth Enterprise Market from 2018 to 2022, a correlation coefficient matrix heatmap of variables was plotted.

In Fig 3, it can be observed that there are strong correlations between the revenue growth rate (*RGR)* and gross revenue growth rate (*GRGR*) (correlation coefficient of 0.96); the net profit growth rate (*NPGR*) and gross profit growth rate (*GPGR*) (correlation coefficient of 0.82); the accounts receivable growth rate (*ARGR*) and gross revenue growth rate (*GRGR*) (correlation coefficient of 0.46); audit opinion (*AO*) and profit health (*PH*) (correlation coefficient of 0.49); and changes in the number of accounting firm engagements (*NCAF*) and *PH* (correlation coefficient of 0.30). The correlation coefficients between the other variables are relatively small, indicating weak linear relationships. Moreover, examining the correlations of each variable suggested that all the financial indicators are mildly negatively correlated with PH, meaning that an increase in these financial indicators leads to a decrease in *PH*, which is favourable for profit health. Conversely, the non-financial indicators *AO* and *NCAF* are strongly positively correlated with PH, meaning that a greater number of non-standard audit

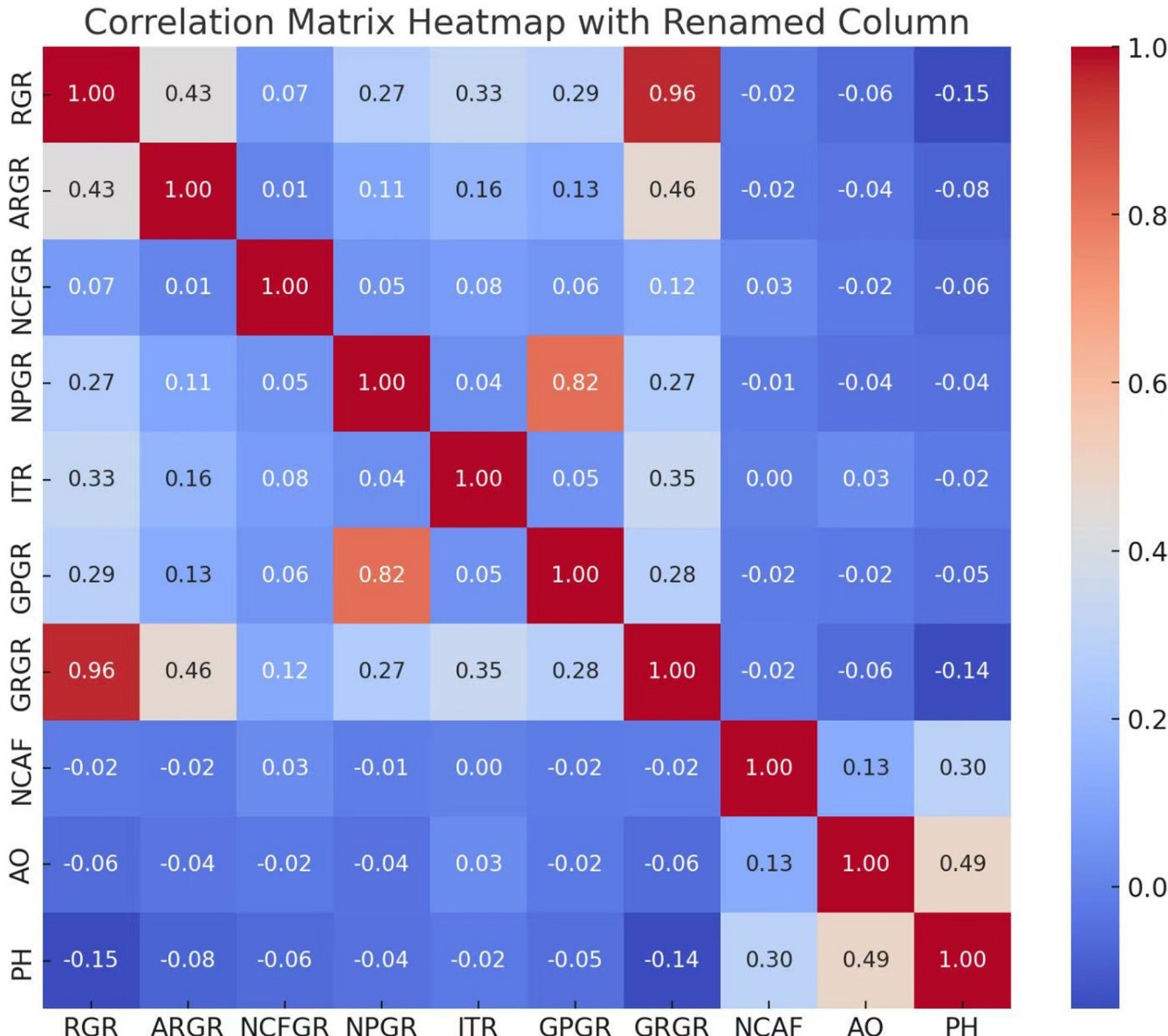

**Fig 3. Correlation matrix heatmap.**

opinions or changes in accounting firm engagements adversely affect *PH*, thereby enabling a preliminary verification that the selected indicators can characterise the degree of profit health of enterprises. Additionally, since some indicators do have strong collinearity, the applicability of the nonlinear regression model test to this article is confirmed.

## 5.5 Model construction

In the measurement model of financial health, the calculation of *PH* is a critical step that determines the accuracy with which the financial health of companies can be determined. Therefore, this study focused on empirically testing hypothesis H and hypotheses $H_1$ to $H_4$.

**5.5.1 Construction of the overall model.** Firstly, employing *ARGR, RGR, NCFGR, ITR, NPGR, GPGR, GRGR, AO*, and *NCAF* as independent variables and profit health as the dependent variable, using the panel data of China's A-share GEM from 2018 to 2022, the gradient boosting machine algorithm (Gradient Boosting Machine, GBM) is used to verify hypothesis *H*.

Given that the GBM is formulated in a step-wise enhancement fashion, it does not possess a straightforward, closed-form functional expression. The dataset was randomly divided into a training set and a test set at an 80–20% ratio, with the learning rate parameter adjusted to 0.2. The dataset was then partitioned into features X and the target variable Y, and the gradient boosting regressor function in Python was employed to construct the GBM model, culminating in the final model as illustrated in Eq (1).

$$F(x) = F_0(x) + \sum_{t=1}^{T} \gamma_t h_t(x) \tag{1}$$

In this equation, $f_0(x)$ signifies the initial estimate of the target; then, for each iteration $t = 1, 2, \ldots, T$, the negative gradient (reflecting the discrepancy between the forecasts of the first $t$-1 models and the actual outcomes) is computed. A novel, less potent prediction model $h_t(x)$ is educated on these residuals, and the optimal step size $\gamma_t$ is ascertained. This step size aims to modulate the contribution of the less potent model.

Following T iterations, the final model comprises the summation of all the less potent models $h_t(x)$ multiplied by $\gamma_t$. It is crucial to acknowledge that the GBM does not manifest as a singular analytical formula; each incremental model introduced per iteration is customarily a decision tree, albeit, in theory, alternate model types could be employed. The GBM's prowess emanates from its capacity to emulate any intricate functional relationship by iteratively refining the loss function.

**5.5.2 Construction of four sub-models.** This study introduced two non-financial indicators, *AO* and *NCAF*, as control variables in each detector to predict *PH*. Based on the random forest algorithm, the explanatory power of each detector for *PH* was examined, thus validating hypotheses $H_1$ to $H_4$.

The formulas for the random forest sub-models of the four detectors are as follows:

$$PH_{ij} = F_1\left(RGR_{ij}, ARGR_{ij}, , AO_{ij}\right), \ i = 2018, 2019, \ldots, 2022; j = 1, 2, \ldots, 580. \tag{I}$$

$$PH_{ij} = F_2\left(RGR_{ij}, NCFGR_{ij}, AO_{ij}\right), \ i = 2018, 2019, \ldots, 2022; j = 1, 2, \ldots, 580. \tag{II}$$

$$PH_{ij} = F_3\left(NPGR_{ij}, ITR_{ij}, AO_{ij}\right), \ i = 2018, 2019, \ldots, 2022; j = 1, 2, \ldots, 580. \tag{III}$$

$$PH_{ij} = F_4\left(GPGR_{ij}, GRGR_{ij}, AO_{ij}\right), \ i = 2018, 2019, \ldots, 2022; j = 1, 2, \ldots, 580. \tag{IV}$$

Through repeated experiments, with the number of trees set to 100, the maximum tree depth set to 10, and the panel data split into a training set (2097 records) and a test set (525 records) using an 80–20% ratio, the models exhibited satisfactory performance.

## 6. Empirical results analysis and robustness test

This section primarily pertains to the examination of two categories of models. Firstly, the gradient boosting machine algorithm is employed to test the overall model, thereby validating the combined impact of nine financial and non-financial indicators on profit health (*PH*).

Secondly, the random forest algorithm is utilized to test four sub-models, with the aim of verifying the influence of a single detector on profit health (*PH*).

## 6.1 Overall model: Empirical results and robustness test

The GBM algorithm is applied to validate the selected indicators, and the main parameters and results are shown in Table 2:

As delineated in Table **2**, all nine independent variables contributed to the dependent variable PH (ascertained by aggregating the scores from four identifiers and two non-financial indicators, with a specific exemplar of a *PH* calculation presented in S1 Appendix). This underscores that the independent variables *RGR, ARGR, NPGR, GPGR, NCFGR, ITR, GRGR*, alongside control variables *AO* and *NCAF*, wield substantial explanatory power over the dependent variable *PH*. Predicated on the varying degrees of variable contributions, *AO* exerts the paramount influence on the entire model, intimating that the audit opinion type predominates when evaluating corporate profit health. Following *AO*, other indicators with notable effects include the inventory turnover rate growth rate (*ITR*), sales revenue growth rate (*RGR*), accounts receivable growth rate (*ARGR*), and total profit growth rate (*GPGR*). These metrics, emblematic of the dynamism of enterprise evolution, elucidate the states of a company's business development, inventory management, and cost control, representing the secondary significant cluster of indicators influencing profit health. Conversely, *NPGR, NCAF, GRGR*, and *NCFGR*—wherein the net profit growth rate and gross revenue growth rate largely stem from other metrics and epitomize passive shifts—are not deemed paramount in impacting profit health. An alteration in accounting firm may hint at profit modifications by the corporation, yet other subjective rationales could also elucidate such shifts. Given the substantial research and development expenditures of ChiNext board entities and their heightened capital requirements, manipulating cash flow presents considerable challenges, thus exerting a minimal influence on profit health. Moreover, the overarching model showcases a diminutive Mean Squared Error (MSE) and an $R^2$ of 0.941, corroborating the model's aptness and predictive prowess, thereby validating hypothesis *H*. (Applying Python gradient boosting machine algorithm modeling and testing, the source program is omitted).

## 6.2 Empirical results and robustness tests of four sub-models

The random forest models corresponding to Detectors $I_1$ to $I_4$ are labelled (*I*) to (*IV*), respectively, with *PH* as the dependent variable in each detector. Each detector includes one pair of financial indicators and two non-financial indicator as explanatory variables. The empirical

**Table 2. Gradient boosting machine algorithm main parameters and results.**

| Main Parameters | Values | Variables | Feature Importance |
|---|---|---|---|
| learning_rate | 0.2 | *AO* | 21.27% |
| max_depth | 5 | *ITR* | 15.29% |
| n_estimators | 150 | *RGR* | 13.64% |
| init | None | *ARGR* | 13.50% |
| subsample | 1:10 | *GPGR* | 12.60% |
| Additional Parameters | Default Settings | *NPGR* | 6.95% |
| *obs* | 2622 | *NCAF* | 6.27% |
| MSE | 0.0916 | *GRGR* | 5.35% |
| $R^2$ | 0.941 | *NCFGR* | 5.14% |

results and robustness tests are shown in Table 3. (Applying Python random forest algorithm to model and test, the source program is omitted).

In Table 3, it can be observed that the $R^2$ of the four sub-models for the training set all exceed 0.6, with a mean absolute error (MAE) of around 0.5. For the test set, the $R^2$ all exceed 0.4 without overfitting (mean squared error, MAE, all below 0.8 or 0.7), indicating that each detector has good explanatory power for *PH* as well as practical value. The feature importance ranking output by each sub-model for the training set shows that except for the relatively small importance of *NCAF*, the other variables have significant impacts on *PH*. Furthermore, it can be observed that in each detector, the dominant indicators affecting *PH* are *AO*, *RGR*, *ITR*, and *GRGR*.

To better characterise the detailed impact of each variable on *PH*, the authors plotted the partial dependence plots of the independent variables and control variables on the *PH* values for each detector to visualise their dynamic evolution. The partial dependence plot of sub-model (*I*), constructed based on Detector $I_1$, is shown in Fig 4. For ease of description, the images in Fig 4 are named *(a)*, *(b)*, *(c)*, and *(d)* from left to right.

Fig 4(A) and 4(B) illustrate that the influence of *RGR* on *PH* marginally surpasses that of ARGR on PH. Nonetheless, both impacts are subordinate to that of *AO* on *PH*, as depicted in Fig 4(D), signifying that *AO* constitutes a principal determinant; this observation aligns with the overall model test outcomes.

Based on Fig 4(A), when the sales revenue growth rate (*RGR*) falls below zero, an increase in *RGR* leads to a rise in the profit health (*PH*) value, signalling an accumulating risk to profit health. In essence, as sales revenue transitions from negative to positive growth, it becomes imperative for investors to closely monitor the profit health level of corporations; when *RGR* exceeds zero, a continual elevation in *RGR* results in a decline in the *PH* value, indicating that escalating sales revenue progressively enhances company profit health. According to Fig 4(B), as the accounts receivable growth rate (*ARGR*) moves from below zero to above, the PH value initially drops and subsequently ascends. This trend intimates that a marginal increase in ARGR might stem from diminished product competitiveness, leading to reduced sales volumes and potentially indicating deteriorated profit health. Once *ARGR* surpasses zero, *PH* generally maintains a low level, suggesting that a high ARGR predominantly favours profit health.

Moreover, the most significant influence on profit health emanates from the audit opinion (*AO*), with non-standard audit opinions elevating the PH value, denoting that the poorer the audit opinion, the less healthy the company's profits appear. For ChiNext privately listed companies, as depicted in Fig 4(D), the effect of changing accounting firms (*NCAF*) on *PH* is relatively minor (likely due to the overall rarity of such occurrences among ChiNext privately listed companies). With an uptick in *NCAF*, there's a gradual increase in the *PH* value,

**Table 3. Empirical results and robustness tests of four random forest sub-models.**

| Category | Key Indicators | Model(*I*) | Model(*II*) | Model(*III*) | Model(*IV*) |
|---|---|---|---|---|---|
| Training set | $R^2$ | 0.6365 | 0.6833 | 0.7341 | 0.7294 |
| | MAE | 0.5884 | 0.5460 | 0.5093 | 0.4932 |
| | MSE | 0.5179 | 0.4512 | 0.3789 | 0.3855 |
| Feature value ranking | | **AO: 32.42%** <br> *RGR*: 30.38% <br> *ARGR*: 27.32% <br> *NCAF*: 9.9% | **RGR: 35.94%** <br> *AO*: 30.26% <br> *NCFGR*: 24.54% <br> *NCAF*: 9.26% | **ITR: 37.16%** <br> *AO*: 27.85% <br> *NPGR*: 26.37% <br> *NCAF*: 8.62% | **GRGR: 39.95%** <br> *AO*: 27.64% <br> *GPGR*: 23.69% <br> *NCAF*: 8.73% |
| Testing set | $R^2$ | 0.4293 | 0.5155 | 0.5699 | 0.5568 |
| | MAE | 0.7593 | 0.6963 | 0.6720 | 0.6564 |
| | MSE | 0.8836 | 0.7501 | 0.6659 | 0.6862 |

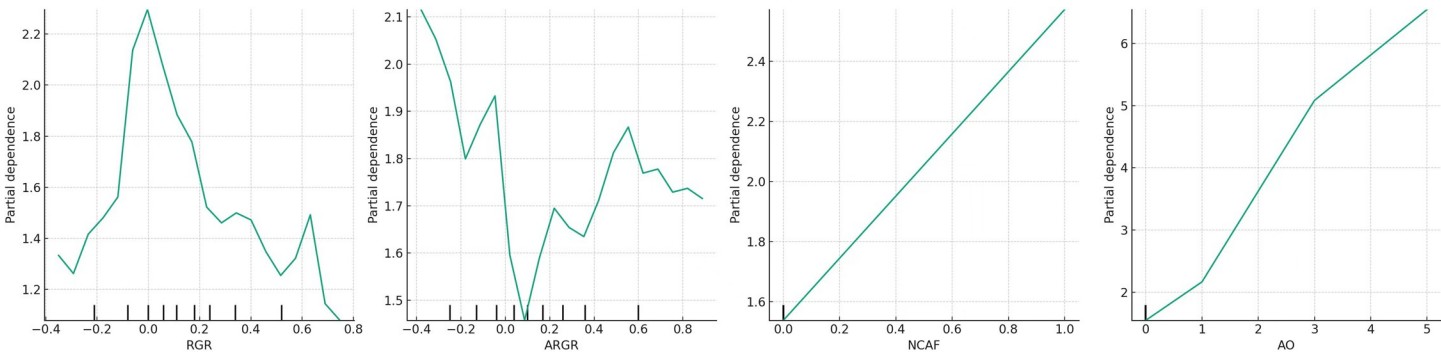

**Fig 4. Partial dependence plot of sub-model (*I*).**

indicating that more frequent changes in accounting firms generally signal less healthy corporate profits.

In summary, the alignment between the accounts receivable growth rate and the sales revenue growth rate indicates favourable profit health, while a misalignment points to poor profit health; thereby, hypothesis $H_1$ is confirmed.

To validate sub-model (*II*) and illustrate the detailed characteristics of the dynamic changes in the internal variables of Detector $I_2$, the partial dependence plot was created, as shown in Fig 5.

For ease of description, the images in Fig 5 are named *(a)*, *(b)*, (c), and *(d)* from left to right.

Referring to Fig 5(C), the behaviour of audit opinion (*AO*) aligns with Identifier 1. Nonetheless, the impact of changing accounting firms (*NCAF*) on *PH*, as shown in Fig 5(D), diverges from Identifier 1, becoming more pronounced. This raises questions: were these accounting firms deemed "incompetent" and thus dismissed by the hiring companies, or did they voluntarily cease auditing these "enigmatic" companies? The truth remains elusive.

As illustrated in Fig 5(A), as *RGR* moves from below zero towards zero, *PH* increases alongside *RGR* growth, signalling a concentration of profit health risk. This may result from a lack of improvement in net cash flow from operating activities, with *NCFGR* still languishing at low levels (as shown in Fig 5(B)).

When *RGR* is above zero, the *PH* value diminishes with *RGR* growth, alleviating profit health risk. This is attributed to the transition of sales revenue growth rate from negative to positive, which can hasten the enhancement of profit health levels. Sustaining positive *RGR* growth continuously ensures the maintenance of profit health.

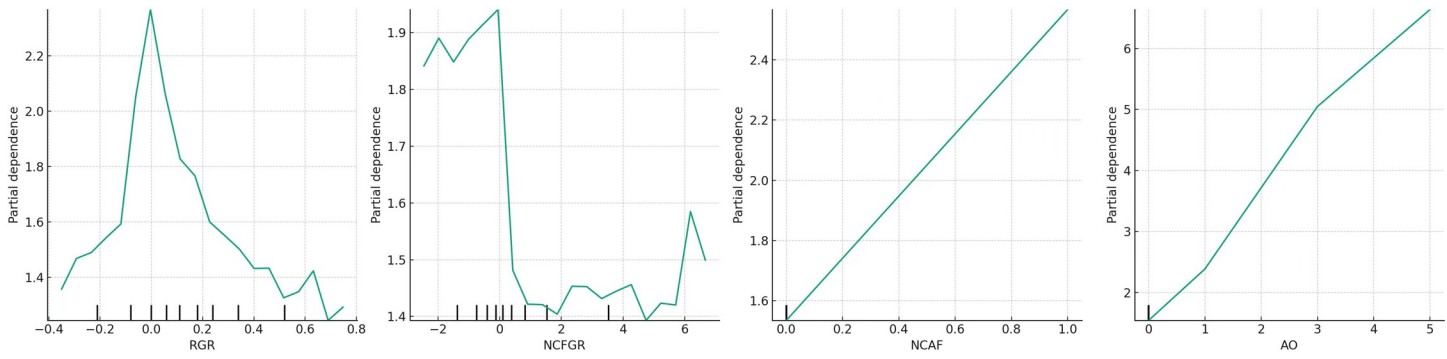

**Fig 5. Partial dependence plot of sub-model (*II*).**

As per Fig 5(B), when the net cash flow from operating activities growth rate (*NCFGR*) is below zero, the profit health risk is elevated, i.e., *PH* is relatively high and slightly increases as *NCFGR* nears a positive figure. This scenario could arise from the sales revenue growth rate significantly outstripping the net cash flow from operating activities. However, when *NCFGR* exceeds zero, it indicates that continual infusions of net cash flow from operating activities foster healthier profits.

In conclusion, if a significant growth in sales revenue is coupled with an increase in the net amount of operating cash flow, it implies healthier company profits. Therefore, their directional changes should generally align; contrasting changes might denote unhealthy profits; thus, $H_2$ is substantiated.

To validate sub-model (*III*) and illustrate the detailed characteristics of the dynamic changes in the internal variables of Detector $I_3$, the partial dependence plot was created, as shown in Fig 6. This figure illustrates that the trends related to *AO* and *NCAF* are consistent with Detector $I_2$. For ease of description, the images in Fig 6 are named *(a)*, *(b)*, *(c)*, and *(d)* from left to right.

According to Fig 6(A), as the net profit growth rate (*NPGR*) falls below zero and approaches zero, the accumulation of profit health risk intensifies. This phenomenon, as depicted in Fig 6(B), relates to an excessively swift growth in inventory turnover, prompting the question of whether companies are intentionally manipulating inventory turnover to the detriment of profit health. Once *NPGR* slightly exceeds zero, the *PH* value undergoes a swift decline (indicating a rapid alleviation of profit health risk) before quickly ascending again, which could suggest that surpassing anticipated net profit growth paradoxically signals unhealthy profits.

From the analysis of Fig 6(B), when the inventory turnover rate growth rate (*ITR*) is negative, profit health levels exhibit significant volatility at a relatively elevated position. Broadly, as the growth rate of inventory turnover decelerates, so too does the level of profit health risk, suggesting an enhancement in inventory management by companies, thereby fostering healthier profits. When *ITR* is positive, the *PH* value first sharply declines and then rapidly increases, settling into a phase of relatively stable fluctuation. This pattern suggests that continual growth in inventory turnover rate is generally advantageous for profit health, albeit excessive growth beyond normal bounds can elevate the *PH* value, potentially undermining profit health.

In summary, stability in inventory turnover rate and net profit growth correlates with healthier corporate profits. Therefore, their directional changes should coincide; divergences may indicate unhealthy profits, thereby validating hypothesis $H_3$.

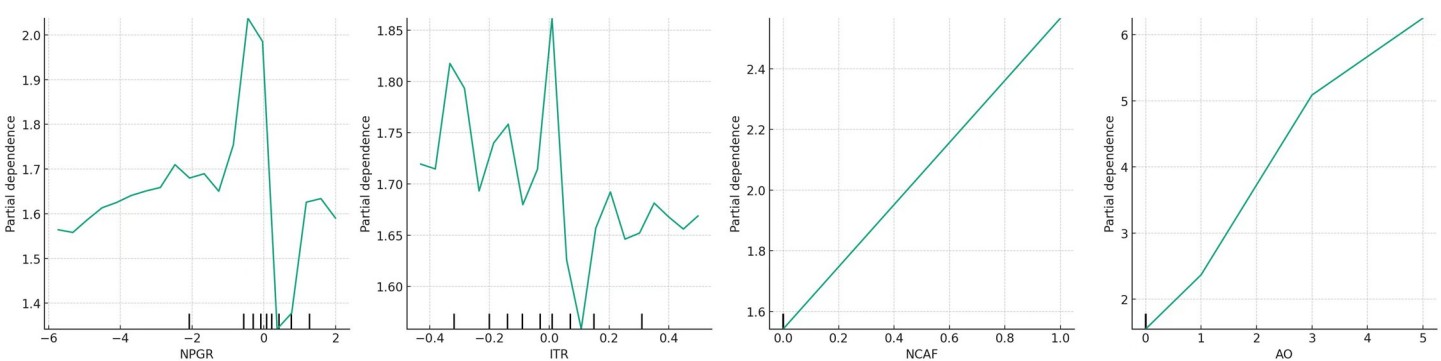

**Fig 6. Partial dependence plot of sub-model (*III*).**

To validate sub-model (*IV*) and illustrate the detailed characteristics of the dynamic changes in the internal variables of Detector $I_2$, the partial dependence plot was created, as shown in Fig 7.

For ease of description, the images in Fig 7 are named *(a)*, *(b)*, *(c)*, and *(d)* from left to right.

Based on Fig 7(B), as the gross revenue growth rate (*GRGR*) swiftly nears and becomes positive around zero, the degree of profit health risk escalates, peaking near zero. Thereafter, as the total profit growth rate (*GPGR*) experiences positive growth, profit health markedly improves, reaching an optimal level as illustrated in Fig 7(A). This outcome signifies that synchronous growth in total profit and gross revenue predicates healthier corporate profits. Conversely, the emergence of an opposite scenario could denote unhealthy profits, thus validating hypothesis $H_4$.

In conclusion, through the visual examination of the four aforementioned identifiers, each variable within every identifier possesses substantial explanatory power regarding profit health levels. Consequently, employing these four identifiers to ascertain the profit health levels of privately listed companies on China's ChiNext board is technically viable.

## 7. Application and expansion

Throughout the development of the gradient boosting machine model and the corroboration of hypotheses $H_1$-$H_4$, the seven financial indicators (*RGR, ARGR, NPGR, GPGR, NCFGR, ITR,* and *GRGR*) and two non-financial indicators (*AO* and *NCAF*) selected for this study have demonstrated significant explanatory capability regarding the determined level of profit health (*PH*). To broaden the notion of profit health to encompass financial health, this paper introduces the classic financial crisis determination model—the Z-score—in an effort to refine the assessment of financial health for listed companies.

### 7.1 Application of profit health evaluation

Employing the profit health assessment framework devised herein and utilizing the panel dataset of privately listed companies on China's ChiNext board from 2018 to 2022 (excluding ST and *ST companies and omitting a few outlier samples), the statistical intervals of the computed PH values are presented in Table 4.

As demonstrated in Table 4, the assessment of corporate profit health is predicated on the mean value of PH and its dispersion across maximum and minimum values, segmented into four uniform intervals as outlined below:

1. Companies with a profit health (*PH*) score ranging from [0 to 4] are deemed to be in a robust profit condition, indicating no evidence of profit manipulation or earnings

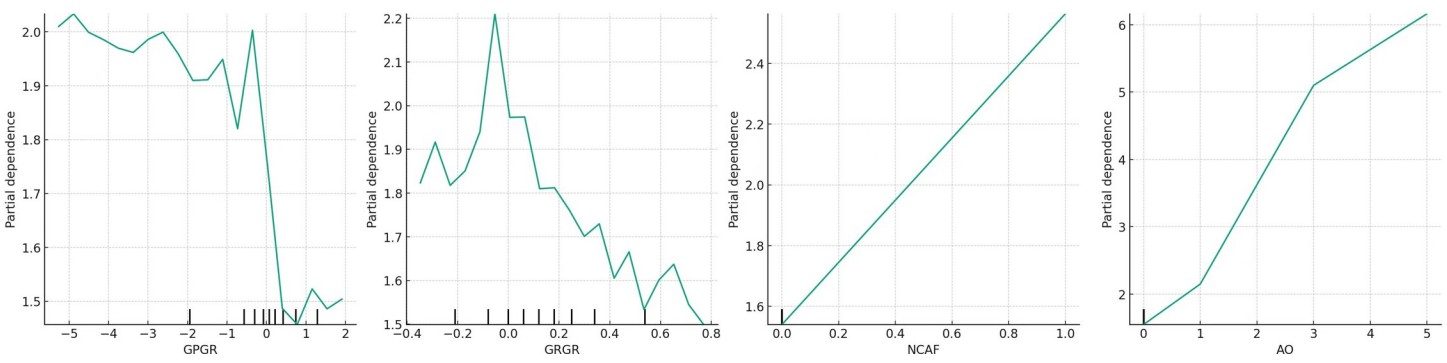

**Fig 7. Partial dependence plot of sub-model (*IV*).**

**Table 4. Profit health conditions of privately listed companies on China's ChiNext Board of A-shares.**

| Variable | Obs | Maximum Value | Minimum Value | Mean Value |
|---|---|---|---|---|
| PH | 503 | 25 | 1 | 8.12 |
| | Range | Health Assessment | Count | Percentage |
| | [0, 4] | Profit Healthy | 45 | 8.95% |
| | [5, 9] | Profit Quasi-healthy | 253 | 50.30% |
| | [10, 17] | Profit Sub-healthy | 197 | 39.17% |
| | [18, +∞) | Profit Unhealthy | 8 | 1.59% |
| Total | | | 503 | 100% |

management, with their profit declarations being reliable and veracious. These enterprises are categorized as "top performers" in profit terms, embodying high investment appeal.

2. Entities possessing a *PH* score between [5 to 9] find themselves in a near-healthy profit condition, implying sporadic, possibly unintentional, profit management activities. The resultant effect on the firm's profit health is negligible, thereby classifying these entities as "good performers," holding substantial investment merit.

3. A *PH* score spanning [10 to 17] signals a company's positioning in a marginal profit health condition, reflective of a likely deliberate participation in earnings management. Such firms are tagged as "under observation" concerning their profit performance, advising investors to tread cautiously and undertake a thorough analysis.

4. Firms registering a *PH* score of [18 to +∞) are perceived to be in a precarious profit state, prone to engage in nefarious profit manipulation or earnings management practices. They are denoted as "poor performers" regarding profit, with the recommendation for investors being to eschew them entirely.

The data reveal that 8.95% of privately owned firms listed on ChiNext fall within the healthy profit bracket, 50.3% within the quasi-healthy profit bracket, 39.17% within the sub-healthy profit bracket, and 1.59% within the unhealthy profit bracket. This illustrates that over half of the companies on ChiNext manifest commendable profit health. The verification confirmed the olive-shaped profit health, as shown in Fig 1.

## 7.2 Application of financial health evaluation

Calculating and determining *PH* can help investors identify the profit health of companies. A higher *PH* indicates a greater likelihood that the management is adjusting earnings or manipulating profits. However, this conclusion cannot provide an overall assessment of the financial condition of companies. Therefore, the grouping results of *PH* were applied to further classify the financial condition of companies in the grey area, with a Z-score model developed to determine the financial condition of companies.

Building upon this foundation, the current study amalgamates the previously explicated *PH* computation and spectrum with the formulation of a Z-score model to refine the financial status delineation of firms within the gray zone. The *PH* score mirrors the robustness of a company's profits and the probability of earnings adjustments. Given that earnings can exert a cumulative influence, prior earnings adjustments may influence subsequent earnings figures, rendering the profit health level (as derived from *PH* values spanning 2018 to 2022) substantively credible. By also considering the present static financial scenario as depicted by the Z-score, a comprehensive evaluation of a company's financial health is achievable. Hence, leveraging the 2022 data, the Z-score for said year was selected for amalgamation with the *PH*

value, enabling the classification of firms' financial health into four distinct conditions: financially healthy, quasi-healthy, sub-healthy, and unhealthy. This classification elucidates the financial health status of companies, as explicated in Table 5.

Table 5 demonstrates that merely 6.76% of privately owned listed companies on the ChiNext board are deemed financially healthy, thereby showcasing significant investment potential. Conversely, 43.34% of the firms are identified as quasi-financially healthy, offering certain investment prospects, while 31.01% are categorized as financially sub-healthy and 18.89% as financially unhealthy, indicating these entities possess low investment allure.

## 7.3 Application extension of financial health evaluation

To verify the cross-sector applicability of the financial health evaluation framework proposed in Table 5, all the privately owned enterprises marked as ST and *ST in China's A-share market in 2022 were exported from the Wind database, along with their financial and non-financial data for 2022. According to the evaluation framework shown in Table 5, the judgement results are shown in Table 6.

Table 6 reveals that all ST and *ST companies are classified under financially sub-healthy or unhealthy segments. Within the context of China's A-share market, ST companies are those flagged for special treatment due to adverse financial conditions or other concerns, such as consecutive losses over two fiscal years or net assets per share falling below share par value in the latest fiscal year. *ST companies, facing imminent delisting risks after persistent losses, must revert to profitability within a set timeframe to evade delisting. The *ST tag, appended to the stock abbreviation, signals elevated risks and delisting warnings from the exchange, marking these firms as bearing severe financial distress and exceedingly high investment hazards, consistent with the financially sub-healthy or unhealthy classifications. Thus, the methodologies and criteria for assessing profit and financial health posited in this study are validated as both theoretically and technically sound.

Given the limited number of ST and *ST companies on the ChiNext (a total of 11), the study's conclusions could bear a bias. To counter this, the analysis was broadened to encompass all privately owned listed entities across the A-share market, involving 75 ST or *ST firms. Their aggregated *PH* scores from 2018 to 2022 were computed according to the financial health assessment framework delineated in Table 5, with 72 companies falling into the financially sub-healthy or unhealthy brackets. Merely two entities were inaccurately classified as financially quasi-healthy, and one as financially healthy, yielding a precision rate of 96%, as detailed in Table 7.

The application results show that the financial health evaluation framework proposed in this study is not only feasible in financial practice but also applicable to the evaluation of the

**Table 5. Financial health conditions of privately owned listed companies on ChiNext.**

| Z score | PH | FH Assessment | Count | Percentage |
|---|---|---|---|---|
| (2.99, +∞) | [0, 4] | Financially Healthy | 34 | 6.76% |
| | [5, 9] | Financially quasi-healthy | 210 | 41.75% |
| | [10, +∞) | Financially sub-healthy | 83 | 16.50% |
| (1.80, 2.99] | [0, 4] | Financially quasi-healthy | 8 | 1.59% |
| | [5, 9] | Financially sub-healthy | 73 | 14.51% |
| | [10, +∞) | Financially unhealthy | 33 | 6.56% |
| (-∞, 1.80] | — | Financially unhealthy | 62 | 12.33% |
| Total | | | 503 | 100% |

**Table 6. Financial health conditions of ST or *ST privately owned listed companies on ChiNext.**

| ID | Security Code | PH | Z-score of 2022 | Financial Health Assessment |
|---|---|---|---|---|
| 1 | 300108.SZ | 18 | -1.69 | Financially unhealthy |
| 2 | 300209.SZ | 17 | -4.98 | Financially unhealthy |
| 3 | 300220.SZ | 11 | 3.60 | Financially sub-healthy |
| 4 | 300282.SZ | 13 | 0.53 | Financially unhealthy |
| 5 | 300301.SZ | 18 | -1.21 | Financially unhealthy |
| 6 | 300313.SZ | 23 | 2.89 | Financially unhealthy |
| 7 | 300427.SZ | 13 | 2.32 | Financially unhealthy |
| 8 | 300495.SZ | 16 | -1.06 | Financially unhealthy |
| 9 | 300555.SZ | 14 | 5.60 | Financially sub-healthy |
| 10 | 300742.SZ | 16 | -1.49 | Financially unhealthy |
| 11 | 300799.SZ | 11 | 80.82 | Financially sub-healthy |

financial health of privately listed companies in different sectors of China's A-share market. To some extent, this validates the cross-sector applicability of the financial health evaluation theoretical model proposed in this study.

## 8. Conclusion

The study initially harnessed the gradient boosting machine (GBM) model for an in-depth exploration of the profit health assessment framework for listed companies, effectively

**Table 7. Financial health determination of ST and *ST privately owned listed companies in China's A-share market.**

| ID | Security Code | Listing Date | PH | Z score of 2022 | Financial Health Assessment |
|---|---|---|---|---|---|
| 1 | ST000023.SZ | 1993/4/29 | 8 | 0.49 | Financially Unhealthy |
| 2 | *ST000046.SZ | 1994/9/12 | 16 | -0.76 | Financially Unhealthy |
| 3 | *ST000416.SZ | 1996/7/19 | 10 | 46.75 | Financially Sub-healthy |
| 4 | ST000525.SZ | 1993/10/28 | 23 | 1.05 | Financially Unhealthy |
| 5 | *ST000996.SZ | 2000/7/18 | 13 | 5.90 | Financially Sub-healthy |
| 6 | ST002002.SZ | 2004/6/25 | 13 | 1.27 | Financially Unhealthy |
| 7 | ST002289.SZ | 2009/9/3 | 10 | 19.29 | Financially Sub-healthy |
| 8 | *ST002699.SZ | 2012/9/11 | 17 | 1.44 | Financially Unhealthy |
| 9 | ST002700.SZ | 2012/9/21 | 15 | 6.91 | Financially Sub-healthy |
| 10 | *ST002776.SZ | 2015/6/26 | 15 | -11.76 | Financially Unhealthy |
| 11 | ST002800.SZ | 2016/5/30 | 10 | 5.08 | Financially Sub-healthy |
| 12 | ST002808.SZ | 2016/8/12 | 11 | 8.98 | Financially Sub-healthy |
| 13 | *ST002816.SZ | 2016/10/25 | 6 | 7.97 | Financially Quasi-healthy |
| 14 | ST002872.SZ | 2017/5/19 | 24 | 1.73 | Financially Unhealthy |
| 15 | ST002951.SZ | 2019/3/15 | 6 | 10.15 | Financially Quasi-healthy |
| 16 | *ST300108.SZ | 2010/8/25 | 21 | -1.69 | Financially Unhealthy |
| 17 | *ST600766.SH | 1996/10/28 | 12 | 3.99 | Financially Sub-healthy |
| 18 | ST600804.SH | 1994/1/3 | 15 | -0.03 | Financially Unhealthy |
| 19 | *ST603133.SH | 2017/3/20 | 14 | 6.31 | Financially Sub-healthy |
| 20 | ST603555.SH | 2014/1/24 | 14 | 2.93 | Financially Unhealthy |
| 21 | ST603557.SH | 2017/8/18 | 16 | -0.30 | Financially Unhealthy |
| ⋮ | ⋮ | ⋮ | ⋮ | ⋮ | ⋮ |
| 75 | *ST688272.SH | 2021/10/18 | 3 | 6.07 | Financially Healthy |

corroborating the selected indicators' substantial explanatory prowess regarding profit health. Among these, the interpretative strength of audit opinions (*AO*) was found to be particularly significant, underscoring the importance of audit opinions in the profit health evaluation of listed companies. Subsequently, each detector was empirically verified one by one using the random forest algorithm, revealing the interaction mechanism between the internal indicators of detectors using the visualisation techniques in machine learning algorithms. Additionally, by combining *PH* with the Z-scores, the framework for estimating the profit health of enterprises can be extended to financial health evaluation. The research's novelty resides in its proposition of an innovative framework for corporate financial health evaluation, adopting a dynamic-static perspective, thereby enhancing the financial crisis prediction models' applicability and scope.

Notably, it articulates a definitive assessment strategy for entities within the financial status "gray zone," furnishing corporate investors and other stakeholders with a pragmatic instrument to deeply understand corporate financial statuses. This methodology additionally facilitates the identification of the most financially robust companies, furnishing a scholarly foundation for investment or business management decision-making. However, it advises a cautious stance towards firms with diminished financial health to avert investment failures.

Nonetheless, this research's framework for financial health assessment does not unequivocally pinpoint specific instances of profit manipulation or earnings management but merely potential financial perils. It also overlooks the comprehensive impact of broader and narrower variables such as sector traits, market dynamics, and policy environments on financial health. Future research should consider including more non-financial factors, such as performance forecasts, relative increases and decreases in the market range, ESG scores, etc., to enrich and improve the financial health assessment theory of listed companies.

## Supporting information

**S1 Appendix. Example calculation of Profit Health (*PH*) value for identifier 1—partial data of privately owned listed companies on China's A-share ChiNext, 2018–2022.** (TIF)

**S1 Dataset.** (XLSX)

## Author Contributions

**Data curation:** Shanqiu Liu.

**Formal analysis:** Wen Zhu.

**Investigation:** Shanqiu Liu.

**Methodology:** Meiling Li.

**Resources:** Chengcheng Wu.

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
