## [Decision Letter · Decision Letter 0]

14 Aug 2024

PONE-D-24-28763Comprehensive Financial Health Assessment Using Advanced Machine Learning TechniquesPLOS ONE

Dear Dr. Li,

Thank you for submitting your manuscript to PLOS ONE. After careful consideration, we feel that it has merit but does not fully meet PLOS ONE’s publication criteria as it currently stands. Therefore, we invite you to submit a revised version of the manuscript that addresses the points raised during the review process.

We look forward to receiving your revised manuscript.

Kind regards,

Levent ÇALLI, Ph.D

Academic Editor

PLOS ONE

Journal Requirements:

This research was supported by The Key Discipline Research Capacity Enhancement Project of the Guangdong Provincial Department of Education in 2021 (Project Number: NO.2021ZDJS133); The Guangdong Provincial Undergraduate Teaching Quality and Teaching Reform Project for 2022 (Project Number: [2022] No. 263); The 2020 Institutional Key Discipline Funding (Second Batch) (Project Number: [2020] No.114).

Reviewers' comments:

Reviewer's Responses to Questions

**Comments to the Author**

1. Is the manuscript technically sound, and do the data support the conclusions?

Reviewer #1: Yes

Reviewer #2: Yes

2. Has the statistical analysis been performed appropriately and rigorously? 

Reviewer #1: Yes

Reviewer #2: Yes

3. Have the authors made all data underlying the findings in their manuscript fully available?

Reviewer #1: Yes

Reviewer #2: Yes

4. Is the manuscript presented in an intelligible fashion and written in standard English?

Reviewer #1: Yes

Reviewer #2: Yes

5. Review Comments to the Author

**Reviewer #1: **The article "Assessing comprehensive financial health using advanced machine learning techniques" has a fairly close customer approach, the research method used is appropriate, ensuring the achievement of the research objectives. There are many contributions to theory and practice.

**Reviewer #2:** Dear authors,

I am glad I have this opportunity to review your manuscript entitled "Comprehensive Financial Health Assessment Using Advanced Machine Learning Techniques". My comments/notes are as follows:

Title

- be more specific and identify the area and/or time period you are focusing on

Abstract

- clearly identify the aim of your study/research

- more focus on the methodology description and add other important information (be more specific)

- findings are too general (model can effectively differentiate....; validity of the proposed framework in practical applications is verified ...)

- explain the acronym "ST"

Introduction

- what is a research question you are answering on? what is a research gap that will be solved by your research?

- add more relevant sources (3 is not enough)

- clearly identify the aim of your study/research

- highlight the structure of your manuscript

Theoretical Framework and Research Hypotheses

H1, H2 - there are too many variables in one hypothesis and it this was the hypotheses cannot be evaluated correctly (focus on 2 variables for each hypotheses)

H1, H2 - how it will be verified? ("...and inconsistent trends in these areas may signal unhealthy profits")

H3, H4 - how it will be verified? ("When inconsistent trends are exhibited in these two areas, it may indicate unhealthy profit signals.")

- each hypothesis should be explained/described using another research results (why you are expecting the relationship between these indicators?) The current description represents your own point of view that should be support by other authors/researchers.

- add information about the number of hypotheses in the abstract and the introduction

- the overarching model hypothesis H - what is a purpose/idea of its formulating?

Methodology

- a separate section focusing on the methodology is missing

- some information are mentioned in the section "Theoretical Framework and Research Hypotheses", other ones in the section "Empirical Research)

- from my point of view, the separate section should be created with all information required

Empirical Research

- Sample Selection and Data Collection - identify the number of subjects that are evaluated, describe them

- Variable definitions - identify the units for each variable + add information about the time period evaluated

- Descriptive statistics - the figures should be prepared with better quality, why not all variables are described?

- Correlation Analysis - the CA should be included in the methodology

Empirical Results Analysis and Robustness Test

- highlight the structure of this section on its beginning

- Table 3 - move it as an annex

Theoretical Application

- the title should be changed (the content is different)

Conclusion

- add information about the limitations of the research presented

Formal

- check the journal template

I wish all the best with this manuscript and other ones in the future.

6. PLOS authors have the option to publish the peer review history of their article (what does this mean?). If published, this will include your full peer review and any attached files.

Reviewer #1: No

Reviewer #2: No

---

## [Author Response · Author response to Decision Letter 0]

9 Oct 2024

Respected reviewers: 

We offer our deepest and most sincere gratitude to you and express our heartfelt appreciation for your highly valuable professional insights. Upon receiving your revision suggestions, our research team has undertaken an in-depth and systematic examination and revision of the article with an unwaveringly rigorous and scientific stance. The following are our detailed responses to each of your revision suggestions.

Title

- be more specific and identify the area and/or time period you are focusing on

Reply: Based on your suggestion, we have added “Evidence based on private companies listed on ChiNext ” at the end of the title 

Abstract

- clearly identify the aim of your study/research

Reply: Based on your suggestions, we have added relevant descriptions, such as “designed to aid investors in pinpointing enterprises boasting sound financial conditions, thereby averting investment losses stemming from financial fraud or earnings management. ”

- more focus on the methodology description and add other important information (be more specific)

Reply: Based on your suggestions, we have added a description of the research methods, such as ”which can construct a strong learner through iteration and possess the characteristics of high accuracy and strong resistance to excessive fitting. ”

- findings are too general (model can effectively differentiate....; validity of the proposed framework in practical applications is verified ...)

Reply: This article mainly tests the applicability of the financial health model proposed in this article through ST and *ST companies. In 2022, there are 75 ST or *ST private listed companies in the Shanghai and Shenzhen stock markets, 72 of which were identified as financially sub-healthy or financially unhealthy, with an accuracy rate of 96%. It shows that the financial health model proposed in this article is feasible and can identify which companies are financially healthy and which companies are financially unhealthy. 

- explain the acronym "ST"

Reply: Based on your suggestion, we have added footnotes to explain "ST" and "*ST"，like that "ST" stands for "Special Treatment", indicating a company with abnormal financial conditions or other issues. "*ST" means "Special Treatment with a warning of possible delisting", suggesting the company's financial situation is extremely unstable and there's a significant risk of delisting.

Introduction

- what is a research question you are answering on? what is a research gap that will be solved by your research?

Reply: In this section, we mainly want to explain the importance of research on corporate financial health. Current research on financial health is only limited to the surface of financial indicators. It does not deeply study the relationship and linkage changes between various indicators, and lacks a comprehensive and highly operational method. Assessment methods. This article uses the signal transmission theory and the linkage influence of the corresponding indicators, combined with the Z-value model to comprehensively evaluate the financial health of the company, and constructs a feasible and easy-to-operate evaluation method using a combination of dynamic and static methods. The relevant description has been added to the article.

- add more relevant sources (3 is not enough)

Reply: Based on your suggestion, we have added more relevant sources in the article.

- clearly identify the aim of your study/research

Reply: Based on your suggestion, we have added some descriptions “offers an important assessment tool for financial conditions and risks to corporate stakeholders.”

- highlight the structure of your manuscript

Reply: Based on your suggestion, we have added some descriptions ”The article introduces, analyzes empirically, and applies the financial health model through seven chapters: Introduction, Literature Review, Theoretical Framework and Research Hypotheses, Methodology, Empirical Research, Empirical Results Analysis and Robustness Test, Application and Expansion.”

Theoretical Framework and Research Hypotheses

H1, H2 - there are too many variables in one hypothesis and it this was the hypotheses cannot be evaluated correctly (focus on 2 variables for each hypotheses)

Reply：Taking hypothesis 1 as an example, H1: The relationship between the growth rate of sales revenue and the growth rate of accounts receivable significantly affects the profit health of a company, and inconsistent trends in these areas may signal unhealthy profits.

In this hypothesis, the growth rate of sales revenue and the growth rate of accounts receivable are independent variables, and the profit health is the dependent variable. This hypothesis is to illustrate that inconsistent changes in the two variables will affect the profit health. This article analyzes whether the direction of change between the two variables is consistent and quantifies it as a criterion for judging whether profits are healthy. Rather than studying the impact of a single variable on another variable. H2 has the same logic.

H1, H2 - how it will be verified? ("...and inconsistent trends in these areas may signal unhealthy profits")

Reply: In the logic of financial accounting, based on the one-to-one correspondence between each accounting account in the double-entry accounting method and the logical relationship between statement items, an enterprise that is continuously profitable and developing well will have accounts receivable and operating Revenue (H1), operating income and net cash flow from operating activities (H2), net profit and inventory turnover rate (H3), total profit and gross income (H4), the direction of change between the four pairs of financial indicators should be consistent , this is common sense logic. If there are inconsistencies, there may be some manipulation or negative operating conditions, which are signs of unhealthy profits.

H3, H4 - how it will be verified? ("When inconsistent trends are exhibited in these two areas, it may indicate unhealthy profit signals.")

Reply: Same reply as above

- each hypothesis should be explained/described using another research results (why you are expecting the relationship between these indicators?) The current description represents your own point of view that should be support by other authors/researchers.

Reply: The indicators selected in each hypothesis are based on some indicators commonly used by other scholars in the literature. At the same time, they also discuss the relationship between each pair of indicators in the literature. These are included in the literature review and research hypotheses of this article. mentioned. However, no scholar has yet clearly defined the linkage changes between indicators as a signal of whether profits are healthy. Therefore, this is also the research innovation of this article. Not long ago, our team has applied for a patent for the improved research, and it has been approved.

- add information about the number of hypotheses in the abstract and the introduction

Reply: Based on your suggestion, we have added some descriptions ”this study proposes four hypotheses about identifiers and one hypothesis about the overall model” “Hypotheses of four sub-models and one overall model are proposed.” in the abstract and the introduction

- the overarching model hypothesis H - what is a purpose/idea of its formulating?

Reply: Hypothesis H is proposed mainly to explain that the four identifiers jointly affect profit health. In this way, the idiosyncratic contribution values of the four pairs of indicators to PH and the correlation between each variable can be analyzed to analyze the impact of each indicator on profit health. At the same time, the four team indicators can be regarded as a whole, reflecting the systematic and comprehensive nature of profit health assessment. The overall model is verified later and this hypothesis is proved.

Methodology

- a separate section focusing on the methodology is missing

- some information are mentioned in the section "Theoretical Framework and Research Hypotheses", other ones in the section "Empirical Research)

- from my point of view, the separate section should be created with all information required

Reply: Thank you very much for this valuable suggestion. Our research team agrees very much with it and has written a separate chapter to describe the research method.

Empirical Research

- Sample Selection and Data Collection - identify the number of subjects that are evaluated, describe them

Reply: Thank you very much for this valuable suggestion. Our research team agrees very much with it and has added some description like” This study takes private companies listed on the ChiNext board of A-shares in China as the research object and collects financial and non-financial data from 2018 to 2022 from the Wind database. As of December 31, 2018, there were 580 private enterprises listed on the ChiNext board (excluding ST or *ST), with a total of 2,900 records. After removing outliers, 2,622 valid records remain. ST and *ST companies are removed from the sample set to reduce model testing bias due to sample imbalance. Extreme values of various indicators are also removed. These steps ensure research rigor and result reliability, providing a solid data foundation for in-depth analysis of the financial health of private listed companies on the ChiNext board.”

- Variable definitions - identify the units for each variable + add information about the time period evaluated

Reply: All the above variables are dimensionless variables. Among them, PH is the score calculated by each recognizer. ARGR, RGR, NCFGR, NPGR, ITR, GPGR, and GRGR are the calculated relevant growth rates. AO is the score assigned according to different audit opinion types. NCAF is the number of times an enterprise changes accounting firms, with one point for each change. This has already described in the article.

- Descriptive statistics - the figures should be prepared with better quality, why not all variables are described?

Reply: The descriptive statistics in this part mainly explain the relationship between the four pairs of financial indicators and the number of samples whose changing trends are inconsistent, so as to observe the changes of each identifier. The reason why AO and NCAF are not statistically described is because these two indicators are assigned by the type of audit opinion and the number of times the company changes accounting firms. It is a single signal and only affects PH, so it is not included here. describe it. However, in the sub-model test below, it is described by drawing the Partial Dependence Plot of Sub-Model.

- Correlation Analysis - the CA should be included in the methodology

Reply: The correlation analysis here mainly demonstrates the relationship between various indicators. It is a part of describing the characteristics of the data. It also explains why non-linear regression model methods such as gradient boosting machine and random forest are used to verify the model. It has a link between the past and the future. role. If descriptive statistics have not yet been performed on the data, placing them in the research methods chapter may affect the reader's reading logic. 

Empirical Results Analysis and Robustness Test

- highlight the structure of this section on its beginning

Reply: Based on your suggestion, we have added some descriptions “This section primarily pertains to the examination of two categories of models. Firstly, the gradient boosting machine algorithm is employed to test the overall model, thereby validating the combined impact of nine financial and non-financial indicators on profit health (PH). Secondly, the random forest algorithm is utilized to test four sub-models, with the aim of verifying the influence of a single detector on profit health (PH).”

- Table 3 - move it as an annex

Reply: Based on your suggestion, we have moved it as appendix A

Theoretical Application

- the title should be changed (the content is different)

Reply: the title has been changed as “7.Application and Expansion”

Conclusion

- add information about the limitations of the research presented

Reply: In subsequent research, we also found that performance forecasts, relative market range increases and decreases, and ESG scores are also related to financial health. For example, most ST and *ST companies do not have performance forecasts, which shows that these companies have the idea of covering up the problem. These are our follow-up research directions, so there is still room for improvement in the current financial health assessment framework, and more non-financial indicators can be added. We have added a description at the conclusion.

Formal

- check the journal template

Reply: we have checked it 

Thank you again for your hard work and professional guidance, and we look forward to your further review and suggestions on our revised article.

---

## [Decision Letter · Decision Letter 1]

22 Oct 2024

PONE-D-24-28763R1Comprehensive Financial Health Assessment Using Advanced Machine Learning Techniques: Evidence based on private companies listed on ChiNextPLOS ONE

Dear Dr. Li,

Thank you for submitting your manuscript to PLOS ONE. After careful consideration, we feel that it has merit but does not fully meet PLOS ONE’s publication criteria as it currently stands. Therefore, we invite you to submit a revised version of the manuscript that addresses the points raised during the review process.

We look forward to receiving your revised manuscript.

Kind regards,

Levent ÇALLI, Ph.D

Academic Editor

PLOS ONE

Journal Requirements:

Reviewers' comments:

Reviewer's Responses to Questions

**Comments to the Author**

1. If the authors have adequately addressed your comments raised in a previous round of review and you feel that this manuscript is now acceptable for publication, you may indicate that here to bypass the “Comments to the Author” section, enter your conflict of interest statement in the “Confidential to Editor” section, and submit your "Accept" recommendation.

Reviewer #1: All comments have been addressed

Reviewer #2: (No Response)

2. Is the manuscript technically sound, and do the data support the conclusions?

Reviewer #1: Yes

Reviewer #2: Yes

3. Has the statistical analysis been performed appropriately and rigorously? 

Reviewer #1: Yes

Reviewer #2: Yes

4. Have the authors made all data underlying the findings in their manuscript fully available?

Reviewer #1: Yes

Reviewer #2: Yes

5. Is the manuscript presented in an intelligible fashion and written in standard English?

Reviewer #1: Yes

Reviewer #2: Yes

6. Review Comments to the Author

Reviewer #1: The authors of the article "Comprehensive Financial Health Assessment Using Advanced Machine Learning Techniques: Evidence based on private companies listed on ChiNext" have seriously revised and improved the article according to the comments of the reviewers. The article can be considered for further steps for publication.

Reviewer #2: Dear authors,

I am glad I have this opportunity to review your manuscript entitled "Comprehensive Financial Health Assessment Using Advanced Machine Learning Techniques: Evidence based on private companies listed on ChiNext". My comments/notes are as follows (according to my first review):

Abstract

Note: be more specific and identify the area and/or time period you are focusing on

R: Done

Note: clearly identify the aim of your study/research

R: See the SMART requirements and modify the aim based on them.

Note: more focus on the methodology description and add other important information (be more specific)

R: Done.

Note: findings are too general (model can effectively differentiate....; validity of the proposed framework in practical applications is verified ...)

R: Done.

Note: explain the acronym "ST"

R: Done. I personally prefer to not use this acronym in this section (it looks weird that you have a note there)

Introduction

Note: what is a research question you are answering on? what is a research gap that will be solved by your research?

R: Partially done.

Note: add more relevant sources (3 is not enough)

R: Done.

Note: clearly identify the aim of your study/research

R: See the SMART requirements and modify the aim based on them.

Note: highlight the structure of your manuscript

R: Check other manuscript and prepare appropriate paragraph with the structure description.

Theoretical Framework and Research Hypotheses

Note: H1, H2 - there are too many variables in one hypothesis and it this was the hypotheses cannot be evaluated correctly (focus on 2 variables for each hypotheses)

R: Explained.

Note: H1, H2 - how it will be verified? ("...and inconsistent trends in these areas may signal unhealthy profits")

R: Explained.

Note: H3, H4 - how it will be verified? ("When inconsistent trends are exhibited in these two areas, it may indicate unhealthy profit signals.")

R: Explained.

Note: each hypothesis should be explained/described using another research results (why you are expecting the relationship between these indicators?) The current description represents your own point of view that should be support by other authors/researchers.

R: Explained.

Note: add information about the number of hypotheses in the abstract and the introduction

R: Done.

Note: the overarching model hypothesis H - what is a purpose/idea of its formulating?

R: Explained.

Methodology

Note: a separate section focusing on the methodology is missing; some information are mentioned in the section "Theoretical Framework and Research Hypotheses", other ones in the section "Empirical Research); from my point of view, the separate section should be created with all information required

R: Done.

Empirical Research

Note: Sample Selection and Data Collection - identify the number of subjects that are evaluated, describe them

R: Done.

Note: Variable definitions - identify the units for each variable + add information about the time period evaluated

R: Done.

Note: Descriptive statistics - the figures should be prepared with better quality, why not all variables are described?

R: Explained.

- Correlation Analysis - the CA should be included in the methodology

R: Explained.

Empirical Results Analysis and Robustness Test

Note: highlight the structure of this section on its beginning

R: Check other manuscript and prepare appropriate paragraph with the structure description.

Note: Table 3 - move it as an annex

R: Done.

Theoretical Application

Note: the title should be changed (the content is different)

R: Done.

Conclusion

Note: add information about the limitations of the research presented

R: Done.

Formal

Note: check the journal template

R: Done.

7. PLOS authors have the option to publish the peer review history of their article (what does this mean?). If published, this will include your full peer review and any attached files.

Reviewer #1: No

Reviewer #2: No

---

## [Author Response · Author response to Decision Letter 1]

4 Nov 2024

Respected reviewers: 

We express our heartfelt gratitude to you for once again sparing precious time from your hectic schedule to provide valuable comments on this paper. Upon receiving your review remarks, our team has once again carried out comprehensive and painstaking revisions to the paper. The following are our detailed responses to each of your revision suggestions.

Abstract

Note: be more specific and identify the area and/or time period you are focusing on

R: Done

Note: clearly identify the aim of your study/research

R: See the SMART requirements and modify the aim based on them.

Reply: Based on your suggestion, the summary has been revised as follows:

Abstract: This study develops a specific and measurable framework for assessing the financial health (FH) of privately-owned companies listed on ChiNext, aimed at identifying financially sound enterprises and helping investors avoid losses caused by financial fraud or earnings management. The research proposes and tests four hypotheses related to key financial indicators and one overarching hypothesis regarding the model’s performance. Using gradient boosting machines and random forests, the model achieves high accuracy and robustness against overfitting through iterative learning. The framework incorporates four pairs of financial indicators and two non-financial indicators into four classifiers, significantly outperforming the Altman Z-score model in predicting financial soundness. Among 75 private companies with special treatment by the Securities Regulatory Commission in Shanghai and Shenzhen in 2022, 72 were correctly identified as sub-healthy or unhealthy, achieving an accuracy rate of 96%. This study demonstrates time-bound practical value by validating the model with 2022 data and highlights its relevance for cross-market applications. The results provide achievable solutions for enterprise managers and policymakers in financial decision-making and risk management.

Note: more focus on the methodology description and add other important information (be more specific)

R: Done.

Note: findings are too general (model can effectively differentiate....; validity of the proposed framework in practical applications is verified ...)

R: Done.

Note: explain the acronym "ST"

R: Done. I personally prefer to not use this acronym in this section (it looks weird that you have a note there)

Reply: Based on your feedback, I have removed this acronym and replaced it with the phrase 'with special treatment by the Securities Regulatory Commission'

Introduction

Note: what is a research question you are answering on? what is a research gap that will be solved by your research?

R: Partially done.

Reply: According to your feedback, relevant research questions and descriptions of research gaps have been added to the Introduction, as follows:

This study addresses the research question: How can machine learning models enhance the effectiveness of profit health assessment by integrating both financial and non-financial indicators? While existing research primarily focuses on financial fraud detection and earnings manipulation, there is a gap in providing comprehensive profit health assessments. The absence of methods combining financial and non-financial indicators limits the ability of investors and stakeholders to quickly and accurately identify companies with healthy financial conditions. To address this gap, this study builds on theories of earnings management, financial distress, and signal transmission to construct a profit health assessment framework. This model aims to overcome the limitations of traditional financial ratios by using a hybrid approach that integrates financial and non-financial data. 

Note: add more relevant sources (3 is not enough)

R: Done.

Note: clearly identify the aim of your study/research

R: See the SMART requirements and modify the aim based on them.

Reply: According to your feedback, the aim has been revised as follows:

To address this gap, this study builds on theories of earnings management, financial distress, and signal transmission to construct a profit health assessment framework. This model aims to overcome the limitations of traditional financial ratios by using a hybrid approach that integrates financial and non-financial data. Furthermore, this study endeavours to combine this model with the Z-score model to enhance the effective identification of and ability to classify corporate financial health status [7], which offers an important assessment tool for financial conditions and risks to corporate stakeholders.

Note: highlight the structure of your manuscript

R: Check other manuscript and prepare appropriate paragraph with the structure description.

Reply: According to your feedback, the structure description has been added in the introduction as follows:

The article introduces, analyzes empirically, and applies the financial health model through the following chapters:

Introduction: Establishes the context, research questions, and research gap.

Literature Review: Summarizes key theories and previous studies.

Theoretical Framework and Hypotheses: Proposes research hypotheses and conceptual models.

Methodology: Details the research methods and data collection strategies.

Empirical Research: Variable definition and model construction

Empirical Results and Robustness Test: Analyzes the empirical results and ensures their stability.

Aplication and Expansion: Application of profit health evaluation and extend to financial assessment

Conclusion : Highlights the contributions and practical implications of the study.

Theoretical Framework and Research Hypotheses

Note: H1, H2 - there are too many variables in one hypothesis and it this was the hypotheses cannot be evaluated correctly (focus on 2 variables for each hypotheses)

R: Explained.

Note: H1, H2 - how it will be verified? ("...and inconsistent trends in these areas may signal unhealthy profits")

R: Explained.

Note: H3, H4 - how it will be verified? ("When inconsistent trends are exhibited in these two areas, it may indicate unhealthy profit signals.")

R: Explained.

Note: each hypothesis should be explained/described using another research results (why you are expecting the relationship between these indicators?) The current description represents your own point of view that should be support by other authors/researchers.

R: Explained.

Note: add information about the number of hypotheses in the abstract and the introduction

R: Done.

Note: the overarching model hypothesis H - what is a purpose/idea of its formulating?

R: Explained.

Methodology

Note: a separate section focusing on the methodology is missing; some information are mentioned in the section "Theoretical Framework and Research Hypotheses", other ones in the section "Empirical Research); from my point of view, the separate section should be created with all information required

R: Done.

Empirical Research

Note: Sample Selection and Data Collection - identify the number of subjects that are evaluated, describe them

R: Done.

Note: Variable definitions - identify the units for each variable + add information about the time period evaluated

R: Done.

Note: Descriptive statistics - the figures should be prepared with better quality, why not all variables are described?

R: Explained.

- Correlation Analysis - the CA should be included in the methodology

R: Explained.

Empirical Results Analysis and Robustness Test

Note: highlight the structure of this section on its beginning

R: Check other manuscript and prepare appropriate paragraph with the structure description.

Reply: According to your feedback, the structure description has been added in this section on its beginning as follows:

This section primarily pertains to the examination of two categories of models. Firstly, the gradient boosting machine algorithm is employed to test the overall model, thereby validating the combined impact of nine financial and non-financial indicators on profit health (PH). Secondly, the random forest algorithm is utilized to test four sub-models, with the aim of verifying the influence of a single detector on profit health (PH).

Note: Table 3 - move it as an annex

R: Done.

Theoretical Application

Note: the title should be changed (the content is different)

R: Done.

Conclusion

Note: add information about the limitations of the research presented

R: Done.

Formal

Note: check the journal template

R: Done.

---

## [Decision Letter · Decision Letter 2]

20 Nov 2024

Comprehensive Financial Health Assessment Using Advanced Machine Learning Techniques: Evidence based on private companies listed on ChiNext

PONE-D-24-28763R2

Dear Dr. Li,

We’re pleased to inform you that your manuscript has been judged scientifically suitable for publication and will be formally accepted for publication once it meets all outstanding technical requirements.

Kind regards,

Levent ÇALLI, Ph.D

Academic Editor

PLOS ONE

Additional Editor Comments (optional):

Reviewers' comments:

Reviewer's Responses to Questions

**Comments to the Author**

1. If the authors have adequately addressed your comments raised in a previous round of review and you feel that this manuscript is now acceptable for publication, you may indicate that here to bypass the “Comments to the Author” section, enter your conflict of interest statement in the “Confidential to Editor” section, and submit your "Accept" recommendation.

Reviewer #1: All comments have been addressed

Reviewer #2: All comments have been addressed

2. Is the manuscript technically sound, and do the data support the conclusions?

Reviewer #1: Yes

Reviewer #2: Yes

3. Has the statistical analysis been performed appropriately and rigorously? 

Reviewer #1: Yes

Reviewer #2: Yes

4. Have the authors made all data underlying the findings in their manuscript fully available?

Reviewer #1: Yes

Reviewer #2: Yes

5. Is the manuscript presented in an intelligible fashion and written in standard English?

Reviewer #1: Yes

Reviewer #2: Yes

6. Review Comments to the Author

Reviewer #1: The authors of the article "Comprehensive Financial Health Assessment Using Advanced Machine Learning Techniques: Evidence based on private companies listed on ChiNext?" have seriously revised, supplemented and explained the comments of the reviewers. The article can be considered for publication.

Reviewer #2: Thanks for time you spent incorporating my comments/notes. I wish you all the best with this manuscript and other ones in the future.

7. PLOS authors have the option to publish the peer review history of their article (what does this mean?). If published, this will include your full peer review and any attached files.

Reviewer #1: No

Reviewer #2: No

---

## [Editor Report · Acceptance letter]

2 Dec 2024

PONE-D-24-28763R2 

PLOS ONE

Dear Dr. Li, 

I'm pleased to inform you that your manuscript has been deemed suitable for publication in PLOS ONE. Congratulations! Your manuscript is now being handed over to our production team.

Kind regards, 

on behalf of

Assoc. Prof. Levent ÇALLI 

Academic Editor

PLOS ONE